# A systematic assessment of preclinical multilaboratory studies and a comparison to single laboratory studies

Victoria T Hunniford[1,2], Agnes Grudniewicz[2], Dean A Fergusson[1,3,4,5], Joshua Montroy[1], Emma Grigor[1,3], Casey Lansdell[1], Manoj M Lalu[1,5,6,7,8]*, On behalf of The Canadian Critical Care Translational Biology Group

[1]Clinical Epidemiology Program, Blueprint Translational Research Group, Ottawa Hospital Research Institute, Ottawa, Canada; [2]Telfer School of Management, University of Ottawa, Ottawa, Canada; [3]Faculty of Medicine, University of Ottawa, Ottawa, Canada; [4]Department of Surgery, University of Ottawa, Ottawa, Canada; [5]School of Epidemiology and Public Health, University of Ottawa, Ottawa, Canada; [6]Department of Anesthesiology and Pain Medicine, The Ottawa Hospital, University of Ottawa, Ottawa, Canada; [7]Regenerative Medicine Program, The Ottawa Hospital Research Institute, Ottawa, Canada; [8]Department of Cellular and Molecular Medicine, University of Ottawa, Ottawa, Canada

## Abstract

**Background:** Multicentric approaches are widely used in clinical trials to assess the generalizability of findings, however, they are novel in laboratory-based experimentation. It is unclear how multilaboratory studies may differ in conduct and results from single lab studies. Here, we synthesized the characteristics of these studies and quantitatively compared their outcomes to those generated by single laboratory studies.

**Methods:** MEDLINE and Embase were systematically searched. Screening and data extractions were completed in duplicate by independent reviewers. Multilaboratory studies investigating interventions using in vivo animal models were included. Study characteristics were extracted. Systematic searches were then performed to identify single lab studies matched by intervention and disease. Difference in standardized mean differences (DSMD) was then calculated across studies to assess differences in effect estimates based on study design (>0 indicates larger effects in single lab studies).

**Results:** Sixteen multilaboratory studies met inclusion criteria and were matched to 100 single lab studies. The multicenter study design was applied across a diverse range of diseases, including stroke, traumatic brain injury, myocardial infarction, and diabetes. The median number of centers was four (range 2–6) and the median sample size was 111 (range 23–384) with rodents most frequently used. Multilaboratory studies adhered to practices that reduce the risk of bias significantly more often than single lab studies. Multilaboratory studies also demonstrated significantly smaller effect sizes than single lab studies (DSMD 0.72 [95% confidence interval 0.43–1]).

**Conclusions:** Multilaboratory studies demonstrate trends that have been well recognized in clinical research (i.e. smaller treatment effects with multicentric evaluation and greater rigor in study design). This approach may provide a method to robustly assess interventions and the generalizability of findings between laboratories.

**Funding:** uOttawa Junior Clinical Research Chair; The Ottawa Hospital Anesthesia Alternate Funds Association; Canadian Anesthesia Research Foundation; Government of Ontario Queen Elizabeth II Graduate Scholarship in Science and Technology

*For correspondence: mlalu@toh.ca

Competing interest: The authors declare that no competing interests exist.

**Editor's evaluation**

This study provides new insights into the strengths of multi center laboratory studies in enhancing rigor and possibly more realistic effect sizes. These insights provide potential paths forward for future studies.

## Introduction

The impact of preclinical research using animal models is conditional on its scientific validity, reproducibility, and representation of human physiology and condition (*Landis et al., 2012*; *Chalmers et al., 2014*; *van der Worp et al., 2010*). Improving the quality of the design, conduct, and reporting of preclinical studies may lead to a reduction in research waste (*Chalmers et al., 2014*; *Ioannidis et al., 2014*), as well as increase their utility in informing the development of novel therapies (*Begley and Ellis, 2012*; *Langley et al., 2017*). One method to do so may be the application of multicenter experimentation in preclinical studies. This is analogous to what is done in clinical trials where positive findings from a single center study are usually evaluated and confirmed in a multicenter study (*Chamuleau et al., 2018*; *Bath et al., 2009*; *Bellomo et al., 2009*). Multicenter studies allow for the comparison of effects between centers, which provides insight into the generalizability of effects across institutions (*Cheng et al., 2017*). Thus, they inherently test reproducibility while also increasing efficiency in attaining sufficient sample sizes (*Bath et al., 2009*). In addition, rigorously designed and reported multicenter studies may enhance the confidence in study findings and increase transparency (*Maertens et al., 2017*). This approach has been adopted in other fields such as social and developmental psychology (*Visser et al., 2022*; *Baumeister et al., 2022*).

Multiple calls from the biomedical science community have been made to adopt and apply multicenter study design to preclinical laboratory-based research (*Langley et al., 2017*; *Chamuleau et al., 2018*; *Maertens et al., 2017*; *O'Brien et al., 2013*; *Boltze et al., 2016*; *Dirnagl and Fisher, 2012*). Some recent examples have been published that exemplify the successful implementation of this approach (*Llovera et al., 2015*; *Jones et al., 2015*; *Maysami et al., 2016*). Indeed, multilaboratory studies may offer a method to test issues of reproducibility that have been highlighted by several studies (*Begley and Ellis, 2012*; *Errington et al., 2021b*). As interest in preclinical multilaboratory studies grows, and major funders begin to invest in this approach (*Federal Ministry of Education and Research, 2022*), a systematic evaluation of this method is needed. This will inform and optimize future multicenter preclinical studies by producing a synthesis of current practices and outcomes, while also identifying knowledge gaps and areas for improvement (*Dirnagl and Fisher, 2012*; *Llovera and Liesz, 2016*; *Fernández-Jiménez and Ibanez, 2015*). Moreover, it is currently unknown how the results obtained from a preclinical multilaboratory study compare to a preclinical study conducted in a single laboratory. This comparison is of interest as multiple *clinical* meta-epidemiological studies have shown that single center clinical trials have a higher risk of bias and overestimate treatment effects compared to multicenter trials (i.e. smaller clinical trials at single sites have a higher probability of methodological shortcomings, lower inferential strength, and may provide inaccurately high estimates of treatment effects) (*Unverzagt et al., 2013*; *Dechartres et al., 2011*; *Bafeta et al., 2012*). For this reason, results from single center studies are generally used cautiously for clinical decision-making (*Bellomo et al., 2009*). Currently, there has been no empirical investigation into whether this trend occurs in the preclinical domain. This knowledge would provide greater insight into the potential value of the multilaboratory design in preclinical research.

The first objective of this systematic review was to identify, assess, and synthesize the current preclinical multilaboratory study literature. The second objective was to empirically determine if differences exist in the methodological rigor and effect sizes between single lab and multilaboratory studies.

## Materials and methods
### Synthesis of preclinical multilaboratory studies

This systematic review was reported in accordance with the Preferred Reporting Items for Systematic Review and Meta-Analysis (PRISMA) guidelines (*Moher et al., 2009*; *Page et al., 2020*). A copy

of the PRISMA checklist is provided in the supporting information (*Reporting standard 1*). The protocol was registered with the International Prospective Register of Systematic Reviews (PROSPERO CRD42018093986). All data can be accessed in supplementary files.

## Preclinical multilaboratory eligibility criteria

### Population
The population of interest was preclinical, interventional, multilaboratory, controlled comparison studies. *Preclinical* was defined as research conducted using nonhuman models that involve the evaluation of potential therapeutic interventions of relevance to human health. *Multilaboratory* was defined as cooperative research formally conducted between multiple research centers (sites). Models were limited to in vivo experiments but were not limited by the clinical scope or domain of the preclinical study.

### Intervention, comparators, outcomes
Interventions were restricted to agents with potential effects when considering human health. There were no limitations to the comparator or outcomes of individual studies included.

### Study design
Eligible preclinical studies including in vivo, controlled, interventional studies of randomized and non-randomized designs. In vivo experiments needed to be conducted at two or more independent sites for the study to qualify as multicentric. The sites needed to also share more than just general study objectives to be considered multicentered. Features that met the 'multicenter' criteria included: shared design, specific hypothesis, a priori protocol, animal model, intervention protocol, method of analysis, primary endpoints tested with or without identical measurement apparatuses; separate centers for coordination, protocol development, and data analysis; and study objective, timelines, protocols, and dissemination strategies developed a priori. Veterinary clinical trials, in vitro and ex vivo studies (with no in vivo component), and retrospective data analysis from multiple sites were excluded.

## Preclinical multilaboratory search strategy

The search strategy was developed in collaboration with our institute's information specialist (Risa Shorr MLS, The Ottawa Hospital). Embase (Embase Classic and Embase), and MEDLINE (Ovid MEDLINE Epub Ahead of Print, In-Process & Other Non-Indexed Citations, Ovid Medline Daily and Ovid Medline) were searched (last updated November 25, 2020). A second, independent librarian peer-reviewed the search strategy according to the Peer Review of Electronic Search Strategy (PRESS) framework (*McGowan et al., 2016*). No study scope, date, or language limits were imposed, though all search terms were in the English language. The search strategy is presented in the supporting information (*Supplementary file 1*), as well as the PRESS review (*Supplementary file 2*).

## Preclinical multilaboratory screening and data extraction

The results from the literature search were uploaded to Distiller Systematic Review Software (DistillerSR; Evidence Partners, Ottawa, Canada). DistillerSR is a cloud-based program that facilitates the review process by managing studies through customized screening, auditing, and reporting. Duplicate references were removed, and two reviewers (VTH and CL/EM/JM) independently screened titles and abstracts based on the eligibility criteria. Any disagreements were resolved by consensus. For the second stage of screening, two reviewers (VTH and MML/JM) independently screened the full-text reports of included references based on the eligibility criteria. Disagreements were solved via consensus.

Data were extracted using a standardized extraction form developed in DistillerSR that was piloted by the primary reviewer (VTH) on five studies and revised based on feedback from a senior team member (MML). Qualitative data included characteristics of the studies: publication details (authors, year published, journal), the country(ies) where the study was conducted, sources of funding, the number of centers involved (experimental and non-experimental), the disease model, animal species and sex, treatment/exposure, all study outcomes (primary, secondary, or undefined), the reported results, statements of barriers and facilitators, and statements of recommendations and suggestions

for future testing of the specific therapy being investigated. Quantitative data included the measures of central tendency and dispersion, the sample sizes for the outcome used in the meta-analysis and for the control group, and the total number of animals analyzed. Numerical data were extracted from raw study data or using Engauge Digitizer (version 12.0 *Mitchell et al., 2020*) if data was presented in a graphical format.

### Assessing preclinical multilaboratory study completeness of reporting and risk of bias

Risk of bias and completeness of reporting in the preclinical multilaboratory studies were assessed independently by two reviewers (VTH and MML), and disagreements were resolved via consensus. For both assessments, the main articles along with the supporting information (when provided) were consulted. All randomized, interventional studies were assessed as high, low, or unclear for the 10 domains of bias adapted from the SYRCLE 'Risk of Bias' assessment tool for preclinical in vivo studies (*Hooijmans et al., 2014*). The 'other sources' of risk of bias domain was divided into four sub-domains (funding influences, conflicts of interest, contamination, and unit of analysis errors). An overall 'other' risk of bias assessment was given based on the following: overall high risk of bias if one or more of the four other sources were assessed as high; overall unclear risk of bias if two or more of the four other sources were assessed as unclear (and no high risk); and overall low risk of bias if three of the four other sources were assessed as low (and no high risk).

Completeness of reporting of the multilaboratory studies was assessed using a checklist modified from various sources: consolidated Standards of Reporting Trials (CONSORT *Moher et al., 2010*); the National Institutes of Health (NIH)'s principles and guidelines for reporting preclinical research *Landis et al., 2012*; and the Good Clinical Practice (GCP) Guidance Document: E6(R2) (*Health Canada, 2016*). The checklist is provided in the supporting information (*Appendix 1—table 5*) with details on the sources for each item.

### Effect size comparison between preclinical multilaboratory and single lab studies

We compared the effect sizes of the multilaboratory studies we identified with single lab studies that investigated the same intervention. We only performed comparisons for the multilaboratory studies that evaluated the efficacy of an intervention. Single lab studies for each of the included multilaboratory study comparisons were identified using rapid review methods, which consisted of the search of a single database, and having a single reviewer screen, extract, and appraise studies while an additional reviewer verified study exclusions, extracted data, and appraisals (*Khangura et al., 2012*; *Varker et al., 2015*). The protocol for the effect size comparison was developed a priori and posted on Open Science Framework (https://osf.io/awvs9/).

### Single lab study eligibility criteria

*Population* - We included all animal species used to model the disease of interest in the multilaboratory study. We only included studies modeling the exact human condition/disease of the multilaboratory study, but did not limit this to the method and timing of disease induction, nor by additional co-morbidities modeled in the animals.

*Intervention* – We included studies investigating the same intervention being evaluated in the multilaboratory study.

*Comparator* – We only included studies that had the same comparator to the multilaboratory study in terms of active versus placebo controls.

*Outcome* – We considered only the main outcome that was evaluated in the multilaboratory study.

*Design* – Eligible preclinical studies including in vivo, controlled, interventional studies of randomized, and non-randomized designs.

*Date limitations* – No date limitations were applied.

### Single lab study screening and data extraction

We first searched for preclinical systematic reviews of the therapies being tested and disease modeled in the multilaboratory studies. If no systematic review was identified, we searched for single lab studies through a formal literature search. Search strategies using the eligibility criteria outlined above were

developed with our institute's information specialist. The references of previous single lab studies cited in the multilaboratory studies were retrieved and used to refine the searches. The database MEDLINE (Ovid MEDLINE Epub Ahead of Print, In-Process & Other Non-Indexed Citations, Ovid Medline Daily, and Ovid Medline) was searched from inception (1946). A validated animal filter limited results to animal studies (*Hooijmans et al., 2010*). A single database was used as per rapid review methods (*Haby et al., 2016*).

The results of the searches were uploaded to DistillerSR. Duplicate references were removed, and one reviewer screened titles, abstracts, and full-text based on eligibility criteria. If after the screening, there were greater than 10 eligible single lab studies identified (either through a systematic review or through a literature search), we selected the 10 single lab studies most similar to the respective multi-laboratory study (in terms of animal species, timing and dose of intervention, time of measurement/humane killing, publication year). Given the large number of searches required, we chose to compare a maximum of 10 single lab studies for feasibility reasons. If more than 10 eligible studies were equally similar to the multilaboratory study, then 10 were randomly selected using a random-number generator (https://www.random.org/).

Data from the eligible single lab studies were extracted by one reviewer (VTH) and audited by a second reviewer (JM). Any disagreements were resolved through further discussion. Extracted data included: the first author, year of publication, the quantitative outcome data along with the measures of variation (e.g. means, standard error/deviation, and sample size) for the shared outcome with the multilaboratory study, the animal species and sex, the study sample sizes (intervention and control groups). Numerical data were extracted from reported study data or using Engauge Digitizer if data was presented in a graphical format.

## Data analysis
### Multilaboratory and single lab study quality
Study quality for both multilaboratory and single lab studies was assessed by one reviewer (VTH) and confirmed by a second (JM); disagreements were resolved by consensus (*Jadad et al., 1996*). Specifically, for each study, we evaluated five key practices that are recognized to reduce bias in laboratory experiments: randomization to treatment groups, low risk of bias methods of randomization (*Reynolds and Garvan, 2021*), blinding of personnel (*O'Connor and Sargeant, 2014*), blinding of outcome assessor (*Bello et al., 2014*; *Macleod et al., 2008*), and complete reporting of all outcome data (*Holman et al., 2016*). Each practice was assessed as '0' for not performed/reported, or a '1' for having performed the practice. We used a Mann-Whitney U test to compare the total quality estimates between multilaboratory and single lab studies. We did not individually compare the five key practices between multilaboratory and single lab studies.

## Statistical analysis - effect size comparison
The multilaboratory study's effect size (i.e. treatment effect) of their respective primary/shared outcome was compared to the pooled effect size of the corresponding set of single lab studies. We extracted quantitative outcome effect measures and measures of variation from each single lab study (e.g. means, standard error/deviation, and sample size). We used Engauge Digitizer if data was presented in a graphical format. Summary effect sizes (ES) were calculated for the multilaboratory and single lab studies using Comprehensive Meta-Analysis (version 3; Biostat Inc, USA). The effect size ratio (ESR) for the multilaboratory versus single lab studies was obtained by dividing the single lab summary effect size by the effect size of the corresponding multilaboratory study. This was expressed as a percentage. For each meta-analysis *i*, this was calculated as:

$$ESR_i = \frac{ES\left(single\ lab\ studies\right)_i}{ES\left(multilaboratory\ study\right)_i}$$

This quantified the difference between the effect size of matched multilaboratory and single lab studies, regardless of the metric used to demonstrate the outcome effect. An ESR of 1 indicates no difference, an ESR greater than 1 indicates single lab studies produce a larger summary effect size compared to multilaboratory studies, and an ESR less than one indicates that the single lab studies produce a smaller summary effect size compared to multilaboratory studies. To obtain a more metric-relevant comparison, and for cases where the single lab studies measure the effect of the

same outcome in different ways, we also calculated the difference of standardized mean difference (DSMD). Because all multilaboratory outcomes were continuous, standardized mean differences were calculated using a random effects inverse-variance model and presented with accompanying 95% confidence intervals. Standardized mean differences were used due to the variety of measurement methods reported for the outcomes of interest. We calculated the standardized mean difference for all single lab studies, collective, and for the multilaboratory study. These values, indicated as d, were used to calculate the DSMD as follows:

$$DSMD_i = d\left(single\ lab\ studies\right)_i - d\left(mutlilaboratory\ study\right)_i$$

## Synthesis of preclinical multilaboratory qualitative data and assessments

Descriptive data of the multilaboratory study was synthesized and presented through tabulation of textual elements (*Popay et al., 2006*). A synthesis of any statements and examples pertaining to barriers and facilitators in conducting a multilaboratory study was also performed. Studies were arranged in tables based on study design, basic characteristics, and risk of bias assessments.

## Synthesis of multilaboratory and single lab data

Study quality estimates and effect size comparison assessments from each of the sets of the selected single lab studies and corresponding multilaboratory studies were synthesized and presented independently in a tabular format. Forest plots were used to compare the effect sizes of individual single lab studies with the respective corresponding multilaboratory study.

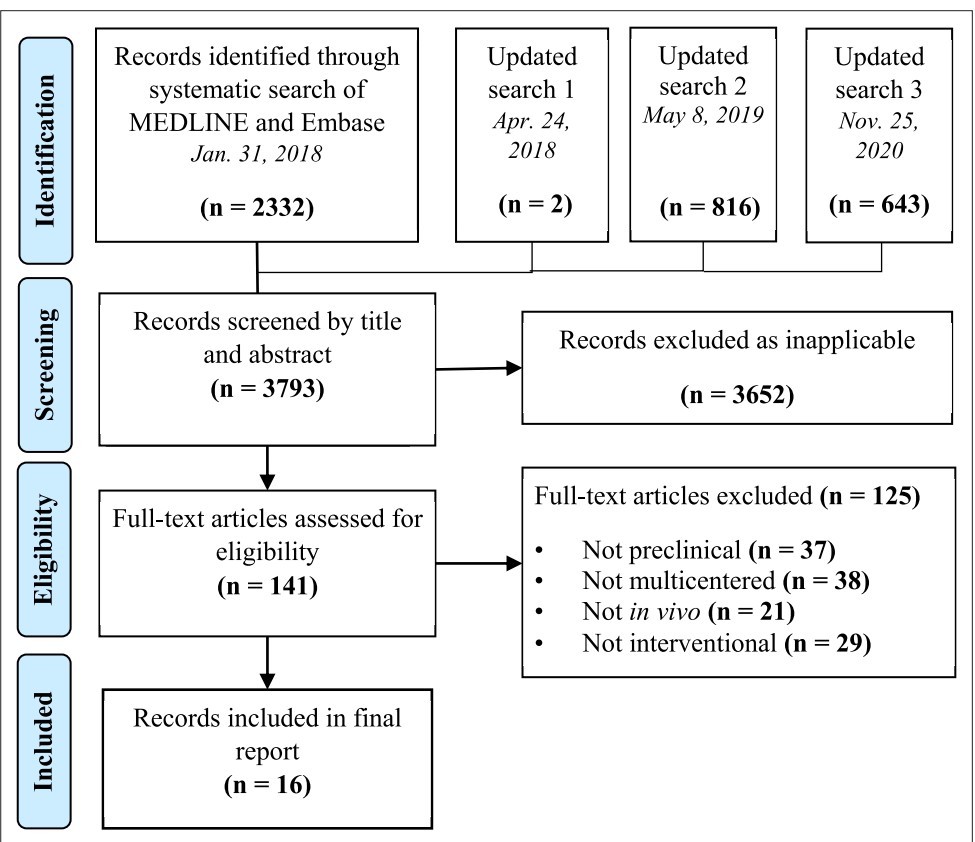

**Figure 1.** Preferred reporting items for systematic reviews and meta-analysis (PRISMA) flow diagram for study selection.

### Deviations from protocol

The original protocol submitted to PROSPERO indicated that the degree of collaboration would be evaluated. After expert feedback, it was decided not to evaluate the degree of collaboration as the methods for this assessment were not feasible to apply to the preclinical setting.

The protocol to evaluate the effect sizes of multilaboratory and single lab studies was posted on Open Science Framework. This protocol indicated that we would use SYRCLE's Risk of Bias tool (*Hooijmans et al., 2014*) to evaluate the studies' *quality estimate*. After expert feedback, it was decided to focus on elements with empirical evidence supporting their importance in the lab setting (i.e. randomization *Reynolds and Garvan, 2021*, blinding *O'Connor and Sargeant, 2014*; *Bello et al., 2014*; *Macleod et al., 2008*, and complete reporting of all outcome data *Holman et al., 2016*).

Peer reviewers requested additional analysis to qualitatively assess identified single laboratory studies that were conducted by the same authors of matched multilaboratory studies.

## Results

### Preclinical multilaboratory search results and study characteristics

The database searches identified a total of 3793 papers after duplicates were removed (*Figure 1*). There were no non-English articles identified in the search. Sixteen articles met eligibility criteria following title, abstract, and full-text screening (*Tables 1 and 2*).

The identified studies fell into seven clinical domains: traumatic brain injury (n=6), myocardial infarction (n=2), stroke (n=2), traumatic injury (n=2), effects of stimulants/neuroactive medication (n=2), diabetes (n=1), and autism spectrum disorder (n=1). Twelve of 16 studies were published in 2015–2020. Five studies were international (studies with labs located in multiple countries), and eleven studies were conducted solely in the USA (all labs located in the USA). The median number of total labs involved per multilaboratory study was four (range: 2–6), and the median number of experimental labs performing in vivo work was three (range:2-5). Nine studies (56%) reported having non-experimental centers involved, such as a coordinating center, data processing center, biomarker core, or pathology core. Five different species of animals were used in the studies: rats (n=7), mice (n=6), swine (n=3), rabbits (n=1), and dogs (n=1). One study used three species of animals for their experiments. The median sample size was 111 (range 23–384 animals), and a total of 2145 animals were used across the sixteen studies, 91% of which were lab rodents (mice and rats).

### Reported preclinical multilaboratory outcomes

Five of the studies (31%) reported that the treatment showed statistically significant, positive results (i.e. favoring the hypothesis); seven studies reported that the treatment showed non-significant or null results; four studies reported that the results were mixed across different treatment specifics (*Alam et al., 2009*), animal models of the disease of interest (*Llovera et al., 2015*), or outcome measures (*Wahlsten et al., 2003*; *Jha et al., 2020*; *Table 2*). Based on their respective results, thirteen studies made explicit statements of recommendations or future directions for the intervention tested. Seven studies stated that they would conduct further testing or recommended that further preclinical testing be done. Four studies indicated they would not continue testing or recommended that no further preclinical testing be done. Three studies recommended to proceed with clinical trials. Brief synopses of the sixteen studies can be found in supporting information (Appendix 1), along with sample statements of their future recommendations (*Appendix 1—table 1*).

### Risk of bias of preclinical multilaboratory studies

None of the 16 studies (0%) were considered low risk of bias across all ten domains (*Table 3*). Fifteen studies randomized animals to experimental groups and four of these reported the method of random sequence generation – one of which used pseudo-randomization methods at one of the participating labs (thus was given a high risk of bias assessment). Thirteen studies had a low risk of detection bias by blinding of outcome assessors. Eleven studies were at low risk of performance bias by blinding personnel administering interventions. All but one study were unclear if animals were randomly housed during the experiments. Six studies from the same research consortium (Operation Brain Trauma Therapy) had a high risk of bias for 'other sources' of bias due to potential industry-related

**Table 1.** Basic study characteristics of preclinical multilaboratory studies.

| Author, Year | Center location | Journal | Funding | Centers Performing In Vivo Work | Non-Experimental Centers* | Animal, Sex | Sample size |
|---|---|---|---|---|---|---|---|
| *Reimer et al., 1985* | US | *Circulation Research* | Government (NHLBI) | 3 | 1 | Dog, both | 51 |
| *Crabbe et al., 1999* | Canada, US | *Science* | Government | 3 | 0 | Mouse, both | 384 |
| *Alam et al., 2009* | US | *Journal of Trauma: Injury, Infection and Critical Care* | Government (US army) | 3 | 0 | Swine, F | 60 |
| *Spoerke et al., 2009* | US | *Archives of Surgery* | Government (US army) | 2 | 0 | Swine, NA | 32 |
| *Jones et al., 2015* | US | *Circulation Research* | Government (NHLBI) | 3 | 3 | Mouse, M<br>Rabbit, M<br>Swine, F | 47<br>23<br>26 |
| *Llovera et al., 2015* | France, Germany, Italy, Spain | *Science Translational Medicine* | Government, academic and charitable | 5 | 1 | Mouse, M | 315 |
| *Maysami et al., 2016* | Finland, France, Germany, Hungary, UK, Spain | *Journal of Cerebral Blood Flow & Metabolism* | Government (FP7/2007–2013, INSERM), academic | 5 | 1[†] | Mouse, M | 241 |
| *Bramlett et al., 2016* | US | *Journal of Neurotrauma* | Government (US army) | 3 | 1 | Rat, M | 140 |
| *Browning et al., 2016* | US | *Journal of Neurotrauma* | Government (US army) | 3 | 1 | Rat, M | 130 |
| *Dixon et al., 2016* | US | *Journal of Neurotrauma* | Government (US army) | 3 | 1 | Rat, M | 135 |
| *Gill et al., 2016* | US | *Diabetes* | Government, charitable | 4 | 0 | Mouse, F | NR |
| *Mountney et al., 2016* | US | *Journal of Neurotrauma* | Government (US army) | 3 | 1 | Rat, M | 128 |
| *Shear et al., 2016* | US | *Journal of Neurotrauma* | Government (US army) | 3 | 1 | Rat, M | 142 |
| *Arroyo-Araujo et al., 2022* | Netherlands, Switzerland, US | *Scientific Reports* | Government (FP7/2007–2013), industry (EFPIA), charitable | 3 | 0 | Rat, M | 72 |
| *Jha et al., 2020* | US | *Journal of Neurotrauma* | Government (US army) | 3 | 1 | Rat, M | 111 |
| *Kliewer et al., 2020* | Australia, Germany, UK | *British Journal of Pharmacology* | Government (NHMRC, NIH), NGO | 3 | 0 | Mouse, M | 108 |

Legend: EFPIA - European Federation of Pharmaceutical Industries and Associations; FP7/2007-2013 – European Union Commission seventh Funding Program; INSERM - Institut national de la santé et de la recherche médicale; NGO – Non-government organization; NHLBI – National Heart, Lung, and Blood Institute; NHMRC – National Health and Medical Research Council; NIH – National Institutes of Health; NR - not reported; UK – United Kingdom; US – United States.

*Non-experimental center: A site/lab not involved with the in vivo experiment (data processing, coordinating, biomarker, or pathology centers).

†Center that was both an experimental center and a coordinating center.

influences (*Table 3*). The four 'other sources' of risk of bias assessments for each study can be found in the supporting information (*Appendix 1—table 2*).

## Completeness of reporting in preclinical multilaboratory studies

Overall, the completeness of reporting of checklist items across all sixteen studies was high (median 72%, range 66–100%). The domains with the highest completeness of reporting included replicates (biological vs. technical), statistics, blinding, and discussion (*Appendix 1—table 3*). The domains

**Table 2.** Study design characteristics of preclinical multilaboratory studies.

| Author, Year | Disease model | Intervention | Study Outcomes | Secondary Outcomes | Reported Results | Recommendations for Future Research |
|---|---|---|---|---|---|---|
| *Reimer et al., 1985* | Myocardial infarction | Verapamil and ibuprofen | Infarct size | Mortality, hemodynamic measures, pathological/ histological features, regional blood flow | Null | Not reported |
| *Crabbe et al., 1999* | Stimulant exposure | Cocaine | Locomotor activity | | Mixed | Further preclinical testing |
| *Alam et al., 2009* | Polytrauma | Blood transfusion | Hemodynamic parameters | Mortality | Mixed across resuscitation products | Further preclinical testing |
| *Spoerke et al., 2009* | Polytrauma | Lyophilized plasma | Residual clotting activity | Mortality, hemodynamic measures, total blood loss, coagulation profiles, inflammatory measures | Positive | Further preclinical testing |
| *Jones et al., 2015* | Myocardial infarction | Ischemic preconditioning | Infarct size | Hemodynamic measures, regional blood flow, heart weight, troponin I, mean arterial pressure | Positive | Further preclinical testing |
| *Llovera et al., 2015* | Stroke | Anti-CD49d antibody | Infarct size | Functional outcome, invasion of leukocytes to brain | Mixed across models (positive, null) | First-in-human clinical trial |
| *Maysami et al., 2016* | Stroke | Interleukin-I receptor antagonist | Infarct size | Odema, functional outcome, mortality | Positive | Extensive clinical trial |
| *Bramlett et al., 2016* | Traumatic brain injury | Erythropoietin | Cognitive outcomes, biomarkers, motor outcomes, neuropathology | | Null | No further preclinical study |
| *Browning et al., 2016* | Traumatic brain injury | Levetiracetam | Cognitive outcomes, biomarkers, motor outcomes, neuropathology | | Positive | Further preclinical testing and first-in-human clinical trial |
| *Dixon et al., 2016* | Traumatic brain injury | Cyclosporine | Cognitive outcomes, biomarkers, motor outcomes, neuropathology | | Null | No further preclinical testing |
| *Gill et al., 2016* | Diabetes | Combined anti-CD3 +IL-1 blockade | Blood glucose | | Null | Pause clinical trial |
| *Mountney et al., 2016* | Traumatic brain injury | Simvastatin | Cognitive outcomes, biomarkers, motor outcomes, neuropathology | | Null | No further preclinical study |
| *Shear et al., 2016* | Traumatic brain injury | Nicotinamide | Cognitive outcomes, biomarkers, motor outcomes, neuropathology | | Null | No further preclinical study |
| *Arroyo-Araujo et al., 2019* | Autism spectrum disorder | mGluR1 antagonist (JNJ16259685) | Behavioural activity | | Positive | Not reported |
| *Jha et al., 2020* | Traumatic brain injury | Glibenclamide | Cognitive outcomes, biomarkers, motor outcomes, neuropathology | Glucose level, drug levels | Mixed across models and outcomes (positive, null, and negative) | Further preclinical testing |
| *Kliewer et al., 2020* | Opioid-induced respiratory depression | Morphine | Respiratory rate | Constipation | Null | Not reported |

**Table 3.** Risk of bias assessment of preclinical multilaboratory studies.

| Study | Sequence generation § | Baseline characteristics | Allocation concealment § | Random housing | Blinding of personnel | Random outcome assessment | Blinding of outcome assessment | Incomplete outcome data *, § | Selective outcome reporting | Other sources of bias ¶ § |
|---|---|---|---|---|---|---|---|---|---|---|
| Reimer et al., 1985 | U* | U | L | U | H† | U | L | U | L | U |
| Crabbe et al., 1999 | U* | L | U | L | U | U | U | L | L | H |
| Alam et al., 2009 | U* | L | U | U | U | L | U | L | L | U |
| Spoerke et al., 2009 | U* | L | U | U | U | L | U | L | L | U |
| Jones et al., 2015 | L | L | U | U | L | L | L | L | L | L |
| Llovera et al., 2015 | L | L | U | U | L | L | L | L | L | L |
| Maysami et al., 2016 | H‡ | H | L | U | L | L | L | L | H | H |
| Bramlett et al., 2016 | U* | U | U | U | L | L | L | U | L | H |
| Browning et al., 2016 | U* | U | U | U | L | L | L | U | L | H |
| Dixon et al., 2016 | U* | U | U | U | L | L | L | U | L | H |
| Gill et al, 2016 | H | H | U | U | U | L | L | U | U | L |
| Mountney et al., 2016 | U* | U | U | U | L | L | L | U | L | H |
| Shear et al., 2016 | U* | U | U | U | L | L | L | U | L | H |
| Arroyo-Araujo et al., 2019 | L | L | U | U | L | U | U | L | L | H |
| Jha et al., 2020 | U* | U | U | U | L | L | L | U | L | H |
| Kliewer et al., 2020 | U* | U | U | U | L | U | L | U | L | L |

*Table 3 continued on next page*

*Table 3 continued*

| Study | Sequence generation § | Baseline characteristics | Allocation concealment§ | Random housing | Blinding of personnel | Random outcome assessment | Blinding of outcome assessment | Incomplete outcome data*, § | Selective outcome reporting | Other sources of bias ¶ § |
|---|---|---|---|---|---|---|---|---|---|---|

Legend: H=High risk of bias (red), L=Low risk of bias (green), U=Unclear risk of bias (yellow).

Baseline Characteristics: Low risk = Relevant baseline characteristics equal between experimental groups or controlled for. Unclear = Relevant baseline characteristics are unreported. High risk = Relevant baseline characteristics unbalanced between experimental groups and not controlled.

Random Housing; Low risk = Animal cages were randomly placed within an animal room/facility, Unclear = Housing placement unreported, High risk = Animals placed in a non-random arrangement in animal room/facility.

Blinding of Outcome Assessment: Low risk = Outcome assessors were blinded to the study groups when assessing endpoints/animals Unclear = Insufficient information to determine if outcome assessors were blinded during an assessment. High Risk = Outcome assessors not blinded to the study groups.

Incomplete Outcome Data: Low risk = N values were consistent between methods and results for the outcomes. Unclear = N value was either not presented in the methods or in the results, and therefore there is insufficient information to permit judgment. High risk = N values were not consistent between methods and results for the outcomes.

Selective Reporting: Low risk = The methods section indicated pre-specified outcome measures. Unclear: Was not clear about the pre-specified primary endpoints and outcome results. High risk = The outcome was presented in the results but not pre-specified in the methods section.

*Method of randomization not specified.

†Assessed as high because one arm of the study was inadvertently unblinded.

‡Some labs used appropriate randomization where others used pseudo-randomization.

§Items in agreement with the Cochrane Risk of Bias tool.

¶other sources include funding influences, conflicts of interest, contamination, a unit of analysis errors.

for standards, randomization, sample size estimation, and inclusion/exclusion criteria were variably reported. The introduction and abstract domain had the lowest completeness of reporting, as 8 of the 16 studies did not report that the study was multicentered in the title (or use a synonym) and less than half indicated the number of participating labs in the abstract. Reporting assessment for all 29 items across the 16 studies can be found in the supporting information (*Supplementary file 3*).

### Single lab study rapid review search results

We next identified single lab studies that were matched to fourteen of the identified multilaboratory studies (these fourteen were used since they evaluated the efficacy of an intervention). Systematic reviews were identified for two interventions (*Wever et al., 2015*; *Peng et al., 2014*), thus systematic searches were designed and undertaken for the remaining twelve studies. In total, 978 articles were screened for eligibility, and data from 100 eligible single lab studies were extracted. Full details of the identification and selection process can be found in Supporting Information, *Appendix 1—table 4*.

### Single lab characteristics

Across the single lab studies, the median number of animals used (number of animals used for all experiments) was 19 (range: 10–72 animals) and the total number of animals used across all studies was 2166. Studies were published between 1980 and 2019. The disease model, treatment, and comparator group were the same in the single lab studies and the respective corresponding multilaboratory studies. Seventy-three percent of the comparisons were made with single lab studies using the same species in the multilaboratory study. Summary characteristics of the included single lab studies are presented in *Table 4*.

### Study quality assessments

Study quality was significantly higher in multilaboratory studies versus single lab studies (p<0.001; Mann-Whitney U test). Across all quality domains, the median score of the multilaboratory studies was assessed as three (range: 1–5), while single lab studies were assessed as two (range: 0–4). Sixty-nine percent of multilaboratory studies compared to 22% of single lab studies had total scores of three and above. Percentage of multi- and single lab studies performing each element assessed are presented in *Table 4*; assessments for each multilaboratory and single lab study can be found in the supplemental information (*Supplementary file 4*).

### Differences in the intervention effect size between single lab and multilaboratory studies

In 13 of 14 comparisons, the intervention effect size (i.e. treatment effect) was larger in single lab studies than in multilaboratory studies. In the pooled analysis of all 14 comparisons, the effect size was significantly larger in single lab studies compared to multilaboratory studies (combined DSMD, 0.72 [95%CI, 0.43–1]; p<0.001) (*Figure 2*). A scatterplot of the study effect sizes for each comparison is presented in *Figure 3*; and the forest plots of each of the 14 comparisons can be found in the supplemental information (*Figure 2—source data 2*). Of note, 8 of the 14 multilaboratory studies had 95% confidence intervals that fell outside of the pooled single laboratory 95% confidence intervals.

### Effect size ratio

The ESR was greater than 1 (i.e. the single lab studies produced a larger summary effect size) in 10 of the 14 comparisons. The median effect size ratio between multilaboratory and single lab studies across all 14 comparisons was 4.18 (range: 0.57–17.14). The ESRs for each comparison along with the mean effect sizes and the ratio of the mean effect sizes are found in the supplemental information (*Supplementary file 4*).

For 10 multilaboratory studies, researchers also had authored 11 matched single lab studies. Median effect size ratio of this smaller matched cohort was 2.50 (mean 4.10, range 0.35–17.63). Median quality assessment score for this cohort of 10 multilaboratory studies was three (range 1–5); median quality score of the 11 single laboratory studies matched by authors was one (range 1–4).

### Reported barriers and enablers to preclinical multilaboratory studies

Five of the 16 studies (31%) explicitly reported on the barriers and facilitators to conducting a multilaboratory study. The most frequently reported barrier identified in all five studies was the establishment

**Table 4.** Comparison of characteristics between single lab and multilaboratory studies.

| | Multilaboratory studies (n=16) | Single lab studies (n=100) |
|---|---|---|
| Median sample size (range) | 111 (23–384) | 19 (10–72) |
| Total animals used | 2,145 | 2,166 |
| Publication date range | 1985–2020 | 1980–2019 |
| Disease model | *n* (%) | *n* (%) |
| TBI | 6 (38) | 46 (46) |
| Myocardial infarction | 2 (13) | 20 (20) |
| Stroke | 2 (13) | 16 (16) |
| Traumatic injury | 2 (13) | 10 (10) |
| Stimulant exposure | 2 (13) | NA |
| Diabetes | 1 (6) | 2 (2) |
| Autism spectrum disorder | 1 (6) | 6 (6) |
| Animal species | *n* (%) | *n* (%) |
| Rat | 7 (44) | 42 (42) |
| Mouse | 6 (38) | 31 (31) |
| Swine | 3 (19) | 12 (12) |
| Rabbit | 1 (6) | 3 (3) |
| Dog | 1 (6) | 9 (9) |
| Monkey | 0 | 2 (2) |
| Cat | 0 | 1 (1) |
| Animal sex | *n* (%) | *n* (%) |
| Male | 10 (63) | 69 (69) |
| Female | 2 (13) | 13 (13) |
| Both | 3 (19) | 12 (12) |
| Not reported | 1 (6) | 6 (6) |
| Quality domain | Percent of studies that performed each measure (%) | |
| Randomization | 94 | 57 |
| Randomization methods | 19 | 7 |
| Blinding of personnel | 69 | 24 |
| Blinding of outcome assessment | 75 | 53 |
| Complete outcome data | 38 | 38 |

of a consistent protocol, with attention to exact experimental details across research labs (*Llovera et al., 2015*; *Jones et al., 2015*; *Maysami et al., 2016*; *Reimer et al., 1985*; *Gill et al., 2016*). In addition to the challenge of the initial protocol development, studies reported difficulty in labs strictly adhering to the established protocol throughout the entirety of the study. One study (*Maysami et al., 2016*) had considerable issues in adhering to the protocol, and in effect had to modify its methods through the course of the study.

Three studies (*Llovera et al., 2015*; *Jones et al., 2015*; *Maysami et al., 2016*) reported differences in equipment and resources across labs as a barrier that made it difficult to conduct a collaborative project and to communicate what measurements and endpoints would be assessed. Specific

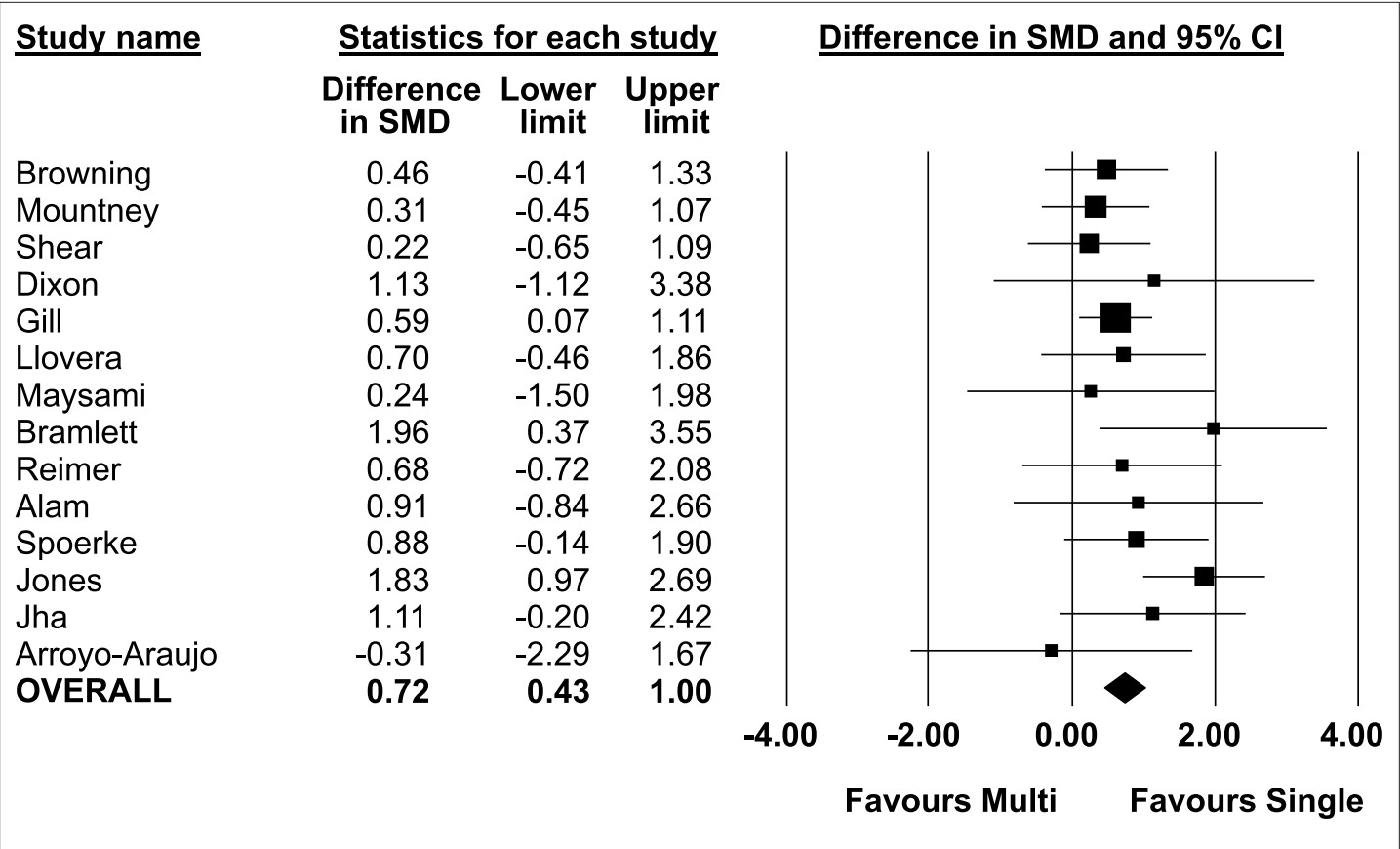

**Figure 2.** Difference in standardized mean difference between single lab and multilaboratory preclinical studies.

The online version of this article includes the following source data for figure 2:

**Source data 1.** Source data for comparision of single lab and multilaboratory studies.

**Source data 2.** Standardized mean differences for all 14 singlevs multilaboratory comparisons and sub-groups by total quality score.

experimental conditions that investigators were unable or unwilling to harmonize across all participating laboratories included animal models of the disease, animal housing conditions, the separate labs' operating and measurement procedures, equipment, and institutional regulations. There was also inconsistent funding across research labs. Different labs had separate budgets with different amounts of funding that could be allocated to the study. If the protocol was to be harmonized, then it had to be adapted to fit each lab's budget accordingly (e.g. the lab with the smallest budget set the spending limit in *Maysami et al., 2016*). Alternatively, labs developed a general protocol but adapted it to fit their own respective budget with what resources they had. Of note, recent work has suggested that harmonization reduces between-lab variability, however, systematic heterogenization did not reduce variability further (*Arroyo-Araujo et al., 2022*); this may suggest that, even in fully harmonized protocols, enough uncontrolled heterogeneity exists that further purposeful heterogenization has little effect. Another barrier identified was ethics approval for animal experiments at all the labs (*Llovera et al., 2015*). This was especially significant when labs were located in multiple countries, as each country had different regulations for ethical approval (*Llovera et al., 2015*; *Maysami et al., 2016*).

*Jones et al., 2015* suggested collaborative protocol development was facilitated by employing pilot testing through all the labs. Developing a defined experimental protocol also included establishing an agreed-upon timeline, laboratory setup, and method of analysis and measurement. *Maysami et al., 2016* and *Reimer et al., 1985* suggested that a similar approach might have enhanced the conduct of both of their studies. Another study reported that the use of a centralized core for administration and data processing was a facilitator (*Llovera et al., 2015*; *Jones et al., 2015*). The validity

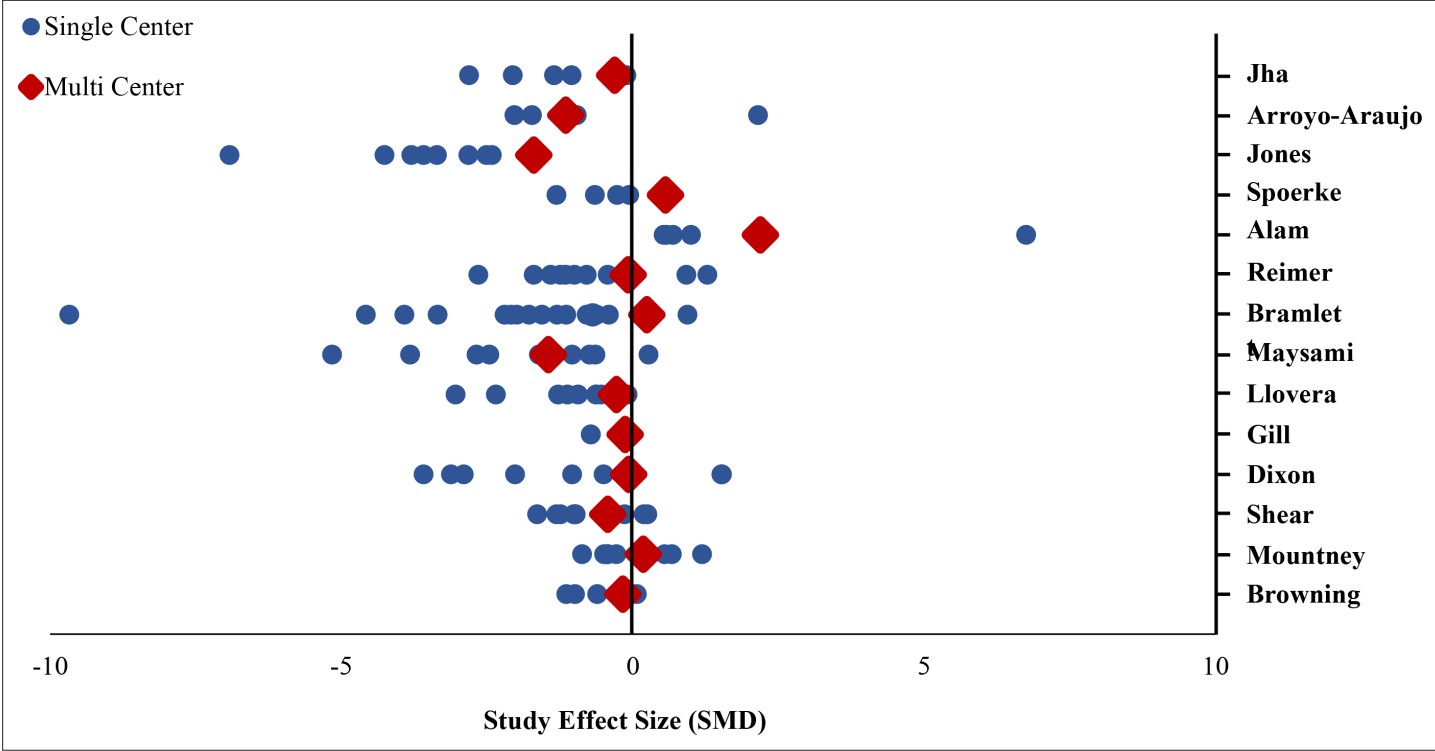

**Figure 3.** Standardized mean differences of single lab and multilaboratory preclinical studies.

of reports depends on the control of statistical and data management, and having one lab coordinate these operations reduces the chances of error or bias in the analysis. Other facilitators were related to the interpersonal aspect of collaboration. These included having investigator leadership through regular conferences and check-ins from the beginning to the end of the project (*Mondello et al., 2016*) and building upon previously established personal/professional relationships between investigators (*Jones et al., 2015*).

## Discussion

Multiple calls for the use of multilaboratory study design in preclinical research have been published (*O'Brien et al., 2013*; *Dirnagl and Fisher, 2012*; *Llovera and Liesz, 2016*; *Fernández-Jiménez and Ibanez, 2015*; *Mondello et al., 2016*). Here, we have synthesized characteristics and outcomes from all interventional preclinical multilaboratory studies published in our search period. Our results suggest that this is an emerging, novel, and promising area of research. The sixteen identified multilaboratory studies had investigated a broad range of diseases, promoted collaboration, adopted many methods to reduce bias, and demonstrated high completeness of reporting. In addition, we found that multi-laboratory studies had higher methodological rigor than single lab studies, demonstrated by the greater level of implementation of several key practices known to reduce bias. We observed that multilaboratory studies showed significantly smaller intervention effect sizes than matched single lab studies. This approach addresses pressing issues that have been recently highlighted such as reproducibility, transparent reporting, and collaboration with data and resource sharing (*Errington et al., 2021b*; *Kane and Kimmelman, 2021*; *Errington et al., 2021a*).

The differences between single and multilaboratory preclinical studies observed here have also been noted in clinical studies. Several comparisons between clinical single-and multilaboratory RCTs have been performed, all finding that single center RCTs demonstrate larger effect sizes (*Bellomo et al., 2009*; *Unverzagt et al., 2013*; *Dechartres et al., 2011*; *Bafeta et al., 2012*). These studies also found that multicenter clinical RCTs had larger sample sizes and greater adherence to practices such as allocation concealment, randomization, and blinding. This difference in sample size and methodological quality, which we have also observed, may explain the discrepancy in effect sizes. It has

been shown that smaller studies included in meta-analyses provide larger intervention effects than larger studies (*Zhang et al., 2013*; *Fraley and Vazire, 2014*). Furthermore, preclinical studies with a high risk of bias (i.e. low methodological quality) may produce inflated estimates of intervention effects (*Landis et al., 2012*; *Collins and Tabak, 2014*). Interestingly, the discrepancy in methodological quality between single and multilaboratory studies was larger in our preclinical comparison than in previous clinical comparisons. This could be explained by the fact that practices such as blinding and randomization are better established in clinical research, thus, even small single center clinical trials are more likely to adhere to them.

A second, somewhat less intuitive issue that may have contributed to larger effect sizes in single lab studies is the smaller sample sizes that may lead to skewed results. It has been suggested that, even in the absence of other biases, under-powered studies have a greater likelihood of effect inflation and generate more false positives than high-powered studies; a lower probability that any observed differences reach the minimum threshold for asserting the findings reflect a true effect; and have a greater likelihood of effect inflation (*Button et al., 2013*). Perhaps the low power of these single lab studies, combined with well-recognized issues of publication bias (*Sena et al., 2010*), contributed to the larger effect sizes we observed.

Completeness of reporting in the multilaboratory studies was also noted to be high across many domains. This is in stark contrast to previous preclinical systematic reviews of single lab studies by our group and others that have found significant deficiencies in reporting (*Landis et al., 2012*; *Fergusson et al., 2019*; *Avey et al., 2016*). Within our sample of multilaboratory studies, replicates, statistics, and blinding were overall transparently reported in the majority of studies. Items specific to multilaboratory designs, such as indicating the number of participating centers in the abstract and identifying as a multilaboratory study in the title were less frequent. One potential explanation for this finding is that guidelines and standards for multilaboratory preclinical studies are just emerging, and there have yet to be any reporting recommendations specific to a preclinical multilaboratory design.

The difference in observed methodological quality and high completeness of reporting in preclinical multilaboratory studies could be explained by the routine oversight and quality control that were employed by some of the multilaboratory studies included in our sample. Though not all multilaboratory studies reported routine oversight, we expect that this is inherent in collaborative studies between multiple independent research groups. As reported by several studies, the coordination of a successful preclinical multilaboratory study requires greater training, standardization of protocols, and study-level management when compared to a preclinical study within a single laboratory. Another barrier was the issue of obtaining adequate funding for a multilaboratory study. As a consequence of limited funding, we would speculate that these studies may have undergone more scrutiny and refinement by multiple investigators, funders, and other stakeholders. Indeed, comparison of single lab studies that had been conducted by authors of multilaboratory studies suggested differences in the conduct and outcomes of these studies (despite having the some of the same researchers involved in both). However, this post hoc analysis was qualitative with a limited sample; thus, future studies will need to explore these issues further.

Due to the greater methodological rigor and transparent reporting, the inferred routine oversight, and larger sample sizes, we speculate that preclinical multilaboratory studies may provide a more precise evaluation of the intervention's effects than do single lab studies. As research groups globally consider adopting this approach, the biomedical community may benefit by emulating successful existing networks of multicenter studies in social psychology (https://psysciacc.org/), developmental psychology (https://manybabies.github.io), and special education research (https://edresearchaccelerator.org/). Moreover, identified barriers and enablers to these studies should be further explored from a variety of stakeholder perspectives (e.g. researchers, animal ethics committees, institutes, and funders) in order to maximize future chances of success.

## Strengths and limitations

A strength of this systematic review is the in-depth synthesis of published preclinical multilaboratory studies that summarizes and assesses the state of this field of research, along with a quantitative comparison between single and multilaboratory studies. The application of rigorous inclusion criteria limited the eligible studies to interventional, controlled-comparison studies, which could omit valuable information that may have come from the excluded studies of non-controlled and/or observational

designs, or studies focused on mechanistic insights. Another limitation is that our assessment of the risk of bias relies on complete reporting; reporting, however, can be influenced by space restrictions in some journals, temporal trends (e.g. better reporting in more recent studies), as well as accepted norms in certain fields of basic science. However, with the increasing prevalence of reporting checklists and standards in preclinical research (*Percie du Sert et al., 2020*), future assessments will be less susceptible to this information bias (*Ramirez et al., 2020*). We also note that our quantitative analysis included only 16 studies, and thus our results might be better regarded as a preliminary analysis that will require future confirmation when more multilaboratory studies have been conducted. We would note, however, that despite the diversity of included multilaboratory studies, overall trends were quite similar across these 16 studies (e.g. more complete reporting, lower risk of bias, and smaller effect size than comparable single laboratory studies).

An additional limitation is that we calculated the effect sizes of the comparable single lab studies using standardized mean differences. We acknowledge that using mean difference would provide a more readily interpretable comparison, however, the use of standardized mean difference allowed us to compare the same outcomes between studies irrespective of the unit of measurement reported. Another limitation is the restriction to a maximum of 10 studies for each multicenter comparison in order to maintain the feasibility of this study. However, we do not expect that this would influence the results or trends we observed.

## Conclusion

This review demonstrates the potential value of multicentric study designs in preclinical research, an approach that has been richly rewarding in clinical research. Importantly, this review provides evidence that preclinical multilaboratory studies report smaller treatment effect sizes and appear to have greater methodological rigor than preclinical studies performed in a single laboratory. This suggests that the preclinical multilaboratory design may have a place in the preclinical research pipeline; indeed, this approach may be a valuable means to evaluate the potential of a promising intervention prior to its consideration in an early-phase clinical trial.

## Acknowledgements

VTH was supported by a Government of Ontario Queen Elizabeth II Graduate Scholarship in Science and Technology. MML was supported by The Ottawa Hospital Anesthesia Alternate Funds Association, a University of Ottawa Junior Clinical Research Chair in Innovative Translational Research, and the Canadian Anesthesiologists' Society Career Scientist Award (Canadian Anesthesia Research Foundation). We would like to thank Risa Shorr (Information Specialist, The Ottawa Hospital) for providing assistance with the generation of the systematic search strategies and article retrieval. We would also like to thank Dr. Alison Fox-Robichaud from the Canadian Critical Care Translational Biology Group for providing critical feedback on the manuscript.

## Additional information

### Funding

| Funder | Grant reference number | Author |
| --- | --- | --- |
| QEII Scholarship | Graduate Student Scholarship | Victoria T Hunniford |
| Canadian Anesthesia Research Foundation | Career Scientist Award | Manoj M Lalu |

The funders had no role in study design, data collection and interpretation, or the decision to submit the work for publication.

### Author contributions

Victoria T Hunniford, Conceptualization, Formal analysis, Investigation, Methodology, Writing – original draft, Writing – review and editing; Agnes Grudniewicz, Supervision, Writing – review and editing;

Dean A Fergusson, Conceptualization, Supervision, Methodology, Writing – review and editing; Joshua Montroy, Formal analysis, Writing – review and editing; Emma Grigor, Casey Lansdell, Investigation, Writing – review and editing; Manoj M Lalu, Conceptualization, Resources, Supervision, Methodology, Writing – review and editing

**Author ORCIDs**
Manoj M Lalu (ID) http://orcid.org/0000-0002-0322-382X

**Decision letter and Author response**
Decision letter https://doi.org/10.7554/eLife.76300.sa1
Author response https://doi.org/10.7554/eLife.76300.sa2

## Additional files

### Supplementary files
- MDAR checklist
- Supplementary file 1. Preclinical multilaboratory search strategy.
- Supplementary file 2. PRESS review of search strategy.
- Supplementary file 3. Completeness of reporting of preclinical multilaboratory studies for 29 reporting items.
- Supplementary file 4. Quality scores, effect sizes, and effect size ratios of multilaboratory and single lab studies.
- Reporting standard 1. PRISMA Checklist.

### Data availability
The protocol for the effect size comparison was developed a priori and posted on Open Science Framework (https://osf.io/awvs9/).Supplementary documents contains the search strategies, risk of bias assessments, reporting checklists, quality scores, effect sizes, effect size ratios, and standardized mean differences to generate the figures and tables.

The following previously published dataset was used:

| Author(s) | Year | Dataset title | Dataset URL | Database and Identifier |
| --- | --- | --- | --- | --- |
| Lalu M, Hunniford V | 2020 | Study Protocol: Quantitative comparison between the effect sizes of preclinical multicenter studies and single center studies | https://osf.io/awvs9/ | Open Science Framework, awvs9 |

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

## Appendix 1

### Supporting information
Overviews of preclinical multicenter studies

1. In a study by *Reimer et al., 1985*, three independent laboratories collaborated to develop models to test potential ischemic myocardium protection therapies, using two standardized, well-characterized canine models of myocardial infarction. Using the two different dog models (conscious model of coronary occlusion, and unconscious model of 3-hr ischemia in open-chest), the researchers tested the effects of verapamil and ibuprofen (therapies) on infarct size. The pooled results from all three centers demonstrated that neither drug limited infarct size in either model. It was later published that the participating laboratories discovered through statistical and hard evidence that a fourth participating lab initially involved in the study had generated fraudulent data, in the sense that data had been completely fabricated by a researcher at one of the centers (*Bailey, 1991*). The data from this lab was not included in the multicenter study paper. The detection of the fraudulent data would not have been possible if not for the design of a multicenter study. The fraud was detected by the large discrepancies in outcome data between the offending center and the other centers involved in the study.

2. *Crabbe et al., 1999* performed a large study across three laboratories. The main objective was to test the behavioural variability in mice of different genetic strains, sexes, and laboratory environments. The evaluation was done with identical methods and protocols across all three labs. The potentially clinically relevant portion of this study was an assessment of cocaine's effect on behavior (i.e. locomotor activity). The study found that cocaine effects on locomotor activity had a strong relationship with genetic differences on the laboratory giving the tests but was negligible for sex differences and source of mice (i.e. shipped from a supplier or bred locally).

3. *Alam et al., 2009* conducted a three-phase severe traumatic injury protocol to model trauma-induced coagulopathy, acidosis, and hypothermia on Yorkshire swine across three experimental centers. Animals were treated with four different blood products: fresh whole blood (FWB), hetastarch, fresh frozen plasma/packed RBCs (FFP: PRBC), and FFP, to determine which, if any, were effective in reversing trauma-associated coagulopathy. Treatment with FFP and FFP: PRBC corrected the coagulopathy as effectively as FWB, whereas hetastarch worsened coagulopathy.

4. *Spoerke et al., 2009* tested whether lyophilized plasma (LP) is as safe and effective as fresh frozen plasma (FFP) for resuscitation after severe trauma. They used a swine model of severe injury across animal laboratories of two level 1 trauma centers, to test the lyophilized plasma for factor levels and clotting activity before lyophilization and after reconstitution. The swine model was developed and performed at one of the centers and was learned and performed at a second center. They found that LP decreased clotting factor activity and was equal to FFP in terms of efficacy.

5. *Jones et al., 2015* aimed to develop a multicenter, randomized controlled clinical-like infrastructure for preclinical evaluation of cardioprotective therapies using mice, rabbit and pig models. The researchers established the Consortium for preclinicAl assESsment of cARdioprotective therapies – called CAESAR - to test the effect of ischemic preconditioning (IPC) on infarct size following a myocardial infarction. IPC involves short episodes of blood restriction to the heart – which is an experimental technique for producing resistance to longer durations of ischemia. Six centers (two centers/animal model) tested the therapy in the three animal models with shared protocols, and found the results were similar across centers and that IPC significantly reduced infarct size in all three species.

6. *Llovera et al., 2015* performed a preclinical randomized controlled multicenter trial to test the potential of anti-CD49d antibodies as a treatment for stroke. These antibodies had shown promise as a form of therapy in individual laboratories by inhibiting the migration of leukocytes into the brain following induction of stroke. Six independent European research centers tested the antibody using two mouse models of stroke. The results demonstrated that the antibody significantly reduced leukocyte invasion and infarct size in the less severe model of stroke. In contrast, these beneficial effects were not noted in the more severe model of stroke.

7. *Maysami et al., 2016* conducted a cross-laboratory study in five centers (four experimental, one coordinating) to test an interleukin receptor antagonist as a drug therapy for stroke. The coordinating center developed and distributed the standard operating procedure to all centers. Stroke

was induced both by permanent and transient occlusion in mice. Drug effects on stroke outcome were evaluated by several measures: lesion volume, edema, neurological deficit scoring and post-treatment mortality. The results across all centers supported the therapeutic potential of the cytokine receptor antagonist in experimental stroke.

8. Six separate studies (2016-2020) that were coordinated by Operation Brain Trauma Therapy (OBTT) consortium (*Kochanek et al., 2016*). Three independent centers collaborated to test 6 different therapies for severe traumatic brain injury (TBI). The consortium was supported by the United States Army and had an overall approach of testing promising therapies in three-well established models of TBI in rats with a rigorous design. The end goal of the consortium was to test the 6 initial therapies in rats prior to considering further testing in a swine model of TBI. Based on the results, four of the six drugs preformed below or well below what was expected based on the previously published literature. It was reported that levetiracetam would advance to testing in the swine model, and that glibenclamide showed benefit only in the cortical contusion injury model.

9. *Gill et al., 2016* assessed the efficacy of combined anti-CD3 plus interleukin-1 blockade to reverse new-onset autoimmune diabetes in non-obese diabetic (NOD) mice. Their consortium was supported by the National Institutes of Health Immune Tolerance Network and the Juvenile Diabetes Research Foundation. Four academic centers shared models and operating procedures. They found that the combined antibody treatment did not show reversal of diabetes across all sites. They did, however, conclude that intercenter reproducibility is possible with the NOD mouse model of diabetes.

10. *Arroyo-Araujo et al., 2019* evaluated the potential of the metabotropic Glutamate Receptor 1 (mGluR1) antagonist JNJ16259685, as a treatment for autism spectrum disorder (ASD). Three centers used *Shank2* knockout (KO) rats as a model for ASD, which mimics autistic-like hyperactivity and repetitive behaviour. They found that the results were reproducible across the three centers, and that KO rats treated with the mGluR1 antagonist demonstrated reduced hyperactivity and repetitive behaviour as compared to placebo treated KO rats.

11. *Kliewer et al., 2020* investigated whether $\beta$-arrestin2 signaling plays a role in opioid-induced respiratory depression. Three independent laboratories injected $\beta$-arrestin2 knockout (KO) mice and control wild-type mice with morphine and monitored the respiratory rate of both groups of mice. The authors found that the KO mice did develop respiratory depression across all three sites, thus, they suggested that β-arrestin2 signaling does not play a key role in opioid-induced respiratory depression.

**Appendix 1—table 1.** Statements of future recommendations.

| Author, Year | Recommendation statements |
| --- | --- |
| *Reimer et al., 1985* | Nothing reported |
| *Crabbe et al., 1999* | *Relatively small genetic effects should first be replicated locally before drawing conclusions... genotypes should be tested in multiple labs and evaluated with multiple tests of a single behavioral domain* |
| *Alam et al., 2009* | *Based upon the findings of the current study that demonstrated the impressive hemostatic properties of plasma, we have proceeded to successfully develop and test (in the same model) lyophilized [freeze dried plasma].* |
| *Spoerke et al., 2009* | *The species-specific differences in factor activities will require ongoing investigation to ensure full safety and efficacy. Our future investigations will include a comprehensive evaluation of the effects of the lyophilization process on coagulation properties of the LP.* |
| *Jones et al., 2015* | *other investigators can adopt the protocols [for measuring infarct size in mice, rabbits, and pigs in a manner that is rigorous, accurate, and reproducible] in their own laboratories.* |
| *Llovera et al., 2015* | *future clinical trials testing immunotherapeutic drugs for stroke will need to ensure that the included study population feature a substantial neuroinflammatory reaction to the brain injury* |

*Appendix 1—table 1 Continued on next page*

*Appendix 1—table 1 Continued*

| Author, Year | Recommendation statements |
|---|---|
| *Maysami et al., 2016* | interleukin 1 receptor antagonist should be evaluated in more extensive clinical stroke trials |
| *Bramlett et al., 2016* | Although we cannot rule out the possibility that other doses or more prolonged treatment could show different effects, the lack of efficacy of EPO reduced enthusiasm for its further investigation in OBTT. |
| *Browning et al., 2016* | …need for OBTT to study LEV further. This includes studies of dose response, therapeutic window, mechanism, and testing in our large animal FPI model in micropigs… consider a randomized controlled trial examining early administration in patients |
| *Dixon et al., 2016* | Our findings reduce enthusiasm for further investigation of this therapy in OBTT and suggest that if this strategy is to be pursued further, alternative CsA analogs with reduced toxicity should be used. |
| *Gill et al., 2016* | …pause in proceeding with clinical trials without further preclinical testing. |
| *Mountney et al., 2016* | the current findings do not support the beneficial effects of simvastatin… it will not be further pursued by OBTT. |
| *Shear et al., 2016* | The marginal benefits achieved with nicotinamide, however, which appeared sporadically across the TBI models, has reduced enthusiasm for further investigation by the OBTT Consortium. |
| *Arroyo-Araujo et al., 2019* | Nothing reported |
| *Jha et al., 2020* | Optimizing [GLY] treatment regimens (dose, duration, timing), surrogate markers for edema subtypes on MRI, pathway-specific biomarkers, and genetic risk stratification may facilitate precision medicine and patient selection for future clinical trials. |
| *Kliewer et al., 2020* | Nothing reported |

Legend: **FDP** – Freeze-dried plasma; **LP** – Lyophilized plasma; **EPO** – Erythropietin; **OBTT** – Operation Brain Trauma Therapy; **LEV** – Levetiracetam; **FPI** – Fluid percussion brain injury; **CsA** – cyclosporin-A; cyclosporine; **TBI** – Traumatic Brain Injury; **GLY** - Glibenclamide.

**Appendix 1—table 2.** Risk of bias for other sources of bias.

| Study | Funding influences | Conflicts of interest | Contamination | Unit of analysis errors |
|---|---|---|---|---|
| *Reimer et al., 1985* | L | U | L | U |
| *Crabbe et al., 1999* | L | U | L | H |
| *Alam et al., 2009* | L | U* | L | U |
| *Spoerke et al., 2009* | L | U* | L | U |
| *Jones et al., 2015* | L | L | L | L |
| *Llovera et al., 2015* | L | L | L | U |
| *Maysami et al., 2016* | L | H | L | U |
| *Bramlett et al., 2016* | L | H | L | U |
| *Browning et al., 2016* | L | H | L | U |
| *Dixon et al., 2016* | L | H | L | U |
| *Gill et al., 2016* | L | L | L | U |
| *Mountney et al., 2016* | L | H | L | U |
| *Shear et al., 2016* | L | H | L | U |

*Appendix 1—table 2 Continued on next page*

*Appendix 1—table 2 Continued*

| Study | Funding influences | Conflicts of interest | Contamination | Unit of analysis errors |
|---|---|---|---|---|
| *Arroyo-Araujo et al., 2019* | H | H | L | L |
| *Jha et al., 2020* | L | H | L | U |
| *Kliewer et al., 2020* | L | L | L | U |

Source of funding: Low risk = Non-industry source of funding/affiliation (or no funding). Unclear = Funding source was not reported. High risk = Study was funded/affiliated by industry.
Conflict of interest: Low risk = Authors reported no conflict of interest. Unclear = Conflict of interest was not reported. High risk = Authors reported on potential conflict of interests.
Contamination: Low risk = No treatment or drug other than the study drug used. Unclear = Possibility of contamination from other treatments or drugs. High risk = Animals receive additional treatment/drugs other than the intervention. Author's report this could influence the results.
Unit of analysis errors: Low risk = Individual units were analyzed individually by the same unit of the treatment comparison group. Unclear: unclear if animals were analyzed individually and treated as one replication. High risk = Units used in the analysis are different from the units of allocation to the treatment comparison groups. Example: animals were not analyzed individual (ex. all animals in one cage) or not treated as one replicate (ex. Same animal: one eye intervention, one eye control).

*financial disclosure, no statement of other conflicts provided.

**Appendix 1—table 3.** Frequency of reported preclinical multilaboratory checklist items.

| Domain | # | Item Description | % of studies that reported |
|---|---|---|---|
| | 1 | Identification as a multicenter/multilaboratory study in title | 38 |
| Intro/ abstract | 2 | Abstract states number of participating centers | 50 |
| | 3 | Community based reporting guidelines listed | 13 |
| | 4 | Names of each participating center listed | 100 |
| | 5 | List roles of participating centers (central coordinating center, experimental site) | 88 |
| Standards | 6 | No changes, or if applicable major changes to study protocol after commencement are documented | 94 |
| | 7 | Results substantiated by repetition under a range of conditions at each site | 100 |
| | 8 | Number of subjects per outcome | 100 |
| | 9 | Number of measurements per subject for one experimental outcome stated | 75 |
| Replicates (biological vs. technical) | 10 | Number of subjects per lab | 81 |
| | 11 | List of the total number of subjects used in each experimental group | 81 |
| | 12 | List of all statistical tests used | 100 |
| | 13 | Definition of the measure of central tendency | 100 |
| Statistics | 14 | Definition of the measure of dispersion | 100 |
| | 15 | Random group assignment reported | 100 |
| Randomization | 16 | Description of the method of random group assignment | 31 |
| | 17 | Experimenters blinded to group allocation during conduct of the experiment | 75 |
| Blinding | 18 | Experimenters blinded to group allocation during result assessment | 75 |

*Appendix 1—table 3 Continued on next page*

*Appendix 1—table 3 Continued*

| Domain | # | Item Description | % of studies that reported |
|---|---|---|---|
| | 19 | Description of an a priori primary outcome | 94 |
| | 20 | Sample size for each site computed during study design | 31 |
| Sample Size Estimation | 21 | Description of the method of sample size determination | 31 |
| | 22 | Total number of animals for the experiment reported | 88 |
| | 23 | Description of the criteria used for the exclusion of any data or subjects | 50 |
| | 24 | List losses and exclusions of animals at the end of experiment | 50 |
| | 25 | All outcomes described, or description of any outcomes that were measured and not reported in the results section | 100 |
| | 26 | Previous or pilot/preliminary studies performed and listed | 88 |
| Inclusion and Exclusion Criteria | 27 | Results were significant, or if not, null or negative outcomes included in the results | 100 |
| | 28 | Limitations of the study are documented | 75 |
| Discussion | 29 | Discrepancies in results across labs expected or absent, or if not, they discussed | 100 |

**Legend:** Coloured cells indicate the frequency (%) of item reported over all included studies. Frequency (%) ranges: 0-37 = red; 38-76 = yellow; 77-100 = green.

**Appendix 1—table 4.** Preclinical single lab studies selection process for the comparison.

| | Reimer et al., 1985 | Spoerke et al., 2009 | Alam et al., 2009 | Llovera et al., 2015 | Maysami et al., 2016 | Gill et al., 2016 | Bramlett et al., 2016 | Browning et al., 2016 | Dixon, 2016 | Mountney et al., 2016 | Shear et al., 2016 | Arroyo-Araujo et al., 2019 | Jha et al., 2020 |
|---|---|---|---|---|---|---|---|---|---|---|---|---|---|
| Records identified/ abstracts screened | 71 | 189 | 61 | 52 | 33 | 39 | 177 | 31 | 197 | 45 | 28 | 26 | 29 |
| Full-texts assessed for eligibility | 29 | 22 | 18 | 14 | 21 | 11 | 57 | 15 | 48 | 30 | 18 | 12 | 10 |
| Records considered eligible | 13 | 5 | 5 | 6 | 10 | 2 | 1 SR | 4 | 8 | 9 | 12 | 6 | 5 |
| Records used for comparison | 10 | 5 | 5 | 6 | 10 | 2 | 1 SR | 4 | 8 | 9 | 10 | 6 | 5 |

**Appendix 1—table 5.** Multilaboratory reporting checklist with item domain and source(s).

| Domain | # | Item Description | Source(s) |
|---|---|---|---|
| Intro/abstract | 1 | Identification as a multicenter/multilaboratory study in title | CONSORT |
| | 2 | Abstract states number of participating centers | CONSORT |
| Standards | 3 | Community based reporting guidelines listed | NIH |
| | 4 | Names of each participating center listed | GCP E6(R2) |
| | 5 | List roles of participating centers (central coordinating center, experimental site) | GCP E6(R2) |
| | 6 | No changes, or if applicable major changes to study protocol after commencement are documented | CONSORT |

*Appendix 1—table 5 Continued on next page*

*Appendix 1—table 5 Continued*

| Domain | # | Item Description | Source(s) |
|---|---|---|---|
| Replicates (biological vs. technical) | 7 | Results substantiated by repetition under a range of conditions at each site | NIH, CONSORT |
| | 8 | Number of subjects per outcome | NIH, CONSORT |
| | 9 | Number of measurements per subject for one experimental outcome stated | NIH, CONSORT |
| | 10 | Number of subjects per lab | GCP E6(R2) |
| Statistics | 11 | List of the total number of subjects used in each experimental group | NIH, CONSORT |
| | 12 | List of all statistical tests used | NIH, CONSORT |
| | 13 | Definition of the measure of central tendency | NIH |
| | 14 | Definition of the measure of dispersion | NIH |
| Randomization | 15 | Random group assignment reported | NIH, CONSORT |
| | 16 | Description of the method of random group assignment | NIH, CONSORT |
| Blinding | 17 | Experimenters blinded to group allocation during conduct of the experiment | NIH, CONSORT |
| | 18 | Experimenters blinded to group allocation during result assessment | NIH, CONSORT |
| Sample Size Estimation | 19 | Description of an a priori primary outcome | CONSORT |
| | 20 | Sample size computed during study design | NIH, CONSORT |
| | 21 | Description of the method of sample size determination | NIH, CONSORT |
| Inclusion and Exclusion Criteria | 22 | Total number of animals for the experiment reported | GCP E6(R2) |
| | 23 | Description of the criteria used for the exclusion of any data or subjects | NIH, CONSORT |
| | 24 | List losses and exclusions of animals at the end of experiment | CONSORT |
| | 25 | All outcomes described, or description of any outcomes measured but not reported in results | NIH, CONSORT |
| | 26 | Previous or pilot/preliminary studies performed and listed | NIH |
| | 27 | Results were significant, or if not, null or negative outcomes included in the results | NIH |
| Discussion | 28 | Limitations of the study are documented | CONSORT |
| | 29 | Discrepancies in results across labs expected or absent, or if not, they discussed | CONSORT |

