## [Editor Report]

This study provides new insights into the strengths of multi center laboratory studies in enhancing rigor and possibly more realistic effect sizes. These insights provide potential paths forward for future studies.

---

## [Decision Letter]

**Decision letter after peer review:**

Thank you for submitting your article "A systematic assessment of preclinical multicenter studies and a comparison to single laboratory studies" for consideration by *eLife*. Your article has been reviewed by 3 peer reviewers, and the evaluation has been overseen by a Reviewing Editor and Mone Zaidi as the Senior Editor. The following individuals involved in the review of your submission have agreed to reveal their identity: David Mellor (Reviewer #1); Andrew SC Rice (Reviewer #3).

Essential revisions:

1) Introduction: I believe that your audience reach would be expanded, and your intended audience would be even more interested in your work, by noting how this method of multi-lab collaborations is appropriate beyond preclinical research. I will not suggest changing your inclusion criteria for the study already conducted, but it would be worth noting how efforts in social psychology* and human development** have seen similar results as those you describe. Likewise, existing networks in psychology (https://psysciacc.org/), development biology (https://manybabies.github.io/), and special education research (https://edresearchaccelerator.org/) would be worth mentioning as models for emulation when discussing how these practices could be scaled to more communities within preclinical research (perhaps mention this in the Conclusion).

2) Line 123: Can you please define "interventional" and clarify your definition of "Preclinical" so as to make it clear to readers outside of your domain of focus why psychological or developmental biology was not included (as these do have human health implications). (Just to be clear, I am not being sarcastic or suggesting that these should have been included, I just know that these terms get blurry when going from the preclinical to the basic sciences).

3) I appreciate that the calculations for the effect size ratio and the difference of standardized mean difference were specified a priori. Even though they were straightforward calculations, there is room for divergence or selective reporting and it appears that those analyses were reported as pre-specified.

4) The sample size is rather limited. While I do not doubt that the authors invested a lot of work into finding multi-centre studies fulfilling their inclusion criteria, they ended up with a list of only 16 studies. Multi-centre studies are tedious to perform and more expensive, so it is not surprising that the authors could not find more. However, given an N=16 for a rather diverse collection of studies, spanning 35 years, different model species (mice, rats, dogs, pigs, and rabbits) and investigating rather different interventions, this might be better regarded as a preliminary analysis.

5) There is one question that cannot be answered but which is nevertheless rather important. Is the difference in effect size estimates (with the multi-lab studies presumably providing more accurate effect size estimates) due to the multi-lab setting itself or due to the fact that researchers willing to go the extra length of conducting a multi-lab study are stricter and more rigorous (and more aware) than the average researcher? If the latter is the case, then this would be a confounding factor and one could not conclude that the observed difference is due to the multi-lab approach itself.

6) As the authors noted themselves, sample sizes of single-lab and multi-lab differed substantially (median 19 vs. 111) and this difference in sample size might itself lead to more accurate effect size estimates (closer to the meta-analytic estimate). I am, therefore, missing an analysis, of whether this effect alone could explain the discrepancy. That is – would single-lab studies using the same number of animals be equally accurate as a multi-lab study?

7) I have some minor methodological issues regarding the evaluation of the risk of bias and specifically its interpretation as a measure for the quality of scientific rigor. This is more of a general critique of the use of 'risk of bias' in systematic reviews and is not specific to this study, but it nevertheless means that the results of this part of the study should be taken with a grain of salt and be interpreted with care. The risk of bias questionnaire is focusing on what is reported in the publication. As it is visible in table 3, even for the multi-lab studies-where risk of bias was considerably lower than for the single-lab studies-a substantial proportion of questions was answered with unclear-i.e., it was, for example, not reported whether allocation concealment was done or not. This could mean that it was not done or it can mean that it was done, but researchers didn't find it noteworthy to report it. Reporting practice slightly differs between fields, it differs between journals (due to space restriction, editorial guidelines, and also the rigor of the editors), and it very strongly changes over time – due to the current discussion about replicability editors and reviewers are now asking for more details than they did a decade ago. So, any analysis of the risk of bias using those criteria should consider journal and year of publication as co-variates. In the present case the authors mentioned, that they tried to match single-lab studies with multi-lab studies also according to age, though a quick look at publication years suggests that this was not always achieved.

8) There seem to be some inconsistencies between Table 3 and Table S11. For example, in the column "Blinding of outcome assessment" in Table S3 there are 4 studies with "unknown" (Crabbe, Alam, Spoerke, Gill), while in Table S11 in the column "Blinding of outcome assessment" there are 4 study scoring 0, but here Arroyo-Araujo scores also 0, while Gill scores 1. How does this fit together?

9) Methods: maybe I have missed that: have you described, whether for the reporting scores you checked only the main article or also the electronic supplementary (supporting information)?

10) Language limits: In the methods description you wrote "No language limits were imposed" (line 147). In the Results section, I could, however, not find any information, on if any non-English papers were found and whether they were included in the analysis. There is also no mention of what you would have done if you encountered a paper in a language you were not capable of. When would this have been an exclusion criterion? In any case, in the methods section where you wrote "no language limits were imposed", please add the information that all search terms were in English. (Or were they also translated into other languages?)

11) Line 166: Primary and secondary research outcomes. According to my own experience, one sees many papers, where the authors report outcomes of several analyses and do not clearly define what the primary and the secondary outcomes are. In fact, I only rarely read that an author writes "this is our primary outcome" and "this was a secondary outcome". Did you encounter such cases and which criteria did you use to decide what counted as a primary and secondary outcome?

12) Line 196: Rapid review method. Please describe in one or two sentences what that is.

13) Line 259: were the effect sizes just pooled or were they weighted by study size or variance?

14) Line 267: I think giving a ratio (without multiplication by 100) might be better

15) Line 345: potentially industry-related influences are considered a risk of bias. Is there actually any study clearly showing that papers with authors from industry are more biased than papers from authors from academia? I know that this is an item on the SYRCLE list of bias tools, but is this prejudice against industry justified – i.e., based on evidence? Just recently there have been several cases of spectacular fraud (data fabrication) by authors in academia (behavioural ecology). Specifically for early career researchers in academia publication pressure for securing tenure or a faculty position is often described as extremely high, creating incentives for "beautifying" results which are then easier to publish. If you are, however, aware of any publication or data showing that bias in industry is more of an issue than in academia then it would be good to cite it here.

16) Line 388: As mentioned in the public review: I would be very careful with equating low reporting scores with "quality of research"

17) Analysis (Meta-analysis, Figure S12): It would be really interesting to add publication year as a covariate in the models. I would assume that estimates get closer to zero the younger the publication.

18) The list of references seems incomplete or incorrect with errors in text citation numbers. There are far more citations in the text (up to 82, whereas only 17 are on the list).

19) It would be good to know more about the training/qualification/validation criteria for the interviewers.

20) page 3 the sentence starting "we found that the…..rigorous manner" did not quite make sense to me.

21) I was curious as to why the group considered that testing of efficacy was the major use of such studies (rather than testing a new mechanism or fundamental finding) and that evaluation of a new method, model, or outcome measure was not even mentioned. That would not have been my view had I been interviewed. It would be nice to see a critical discussion about this.

22) I wonder if a more forensic discussion of what the authors see as the probable barriers to funders, peer scientists, and other stakeholders would be informative. Perhaps even a more extensive discussion of potential solutions may have been informative.

23) The main area of disagreement between interviewees was the issue of harmonisation of protocols- a strong contrast to the clinical multi-centre practice where harmonisation is the norm. This interesting response could perhaps have been discussed more?

Ebersole, C. R., Atherton, O. E., Belanger, A. L., Skulborstad, H. M., Allen, J. M., Banks, J. B., Baranski, E., Bernstein, M. J., Bonfiglio, D. B. V., Boucher, L., Brown, E. R., Budiman, N. I., Cairo, A. H., Capaldi, C. A., Chartier, C. R., Chung, J. M., Cicero, D. C., Coleman, J. A., Conway, J. G., … Nosek, B. A. (2016). Many Labs 3: Evaluating participant pool quality across the academic semester via replication. Journal of Experimental Social Psychology, 67, 68-82. https://doi.org/10.1016/j.jesp.2015.10.012

Klein, R. A., Ratliff, K. A., Vianello, M., Adams, R. B., Bahník, Š., Bernstein, M. J., Bocian, K., Brandt, M. J., Brooks, B., Brumbaugh, C. C., Cemalcilar, Z., Chandler, J., Cheong, W., Davis, W. E., Devos, T., Eisner, M., Frankowska, N., Furrow, D., Galliani, E. M., … Nosek, B. A. (2014). Investigating Variation in Replicability: A "Many Labs" Replication Project. Social Psychology, 45(3), 142-152. https://doi.org/10.1027/1864-9335/a000178

Klein, R. A., Vianello, M., Hasselman, F., Adams, B. G., Adams, R. B., Alper, S., Aveyard, M., Axt, J. R., Babalola, M. T., Bahník, Š., Batra, R., Berkics, M., Bernstein, M. J., Berry, D. R., Bialobrzeska, O., Binan, E. D., Bocian, K., Brandt, M. J., Busching, R., … Nosek, B. A. (2018). Many Labs 2: Investigating Variation in Replicability Across Samples and Settings. Advances in Methods and Practices in Psychological Science, 1(4), 443-490. https://doi.org/10.1177/2515245918810225

Byers-Heinlein, Tsui, A. S. M., K., Bergmann, C., …, Wermelinger, S. (2021). A multi-lab study of bilingual infants: Exploring the preference for infant-directed speech. Advances in Methods and Practices in Psychological Science. PsyArXiv PrePrint

Byers-Heinlein, K., Tsui, R. K. Y., van Renswoude, D., Black, A. K., Barr, R., Brown, A., … Singh, L. (2020). The development of gaze following in monolingual and bilingual infants: A multi-lab study Infancy, Advance online publication. PsyArXiv Preprint

ManyBabies Consortium (2020). Quantifying sources of variability in infancy research using infant-directed speech preference. Advances in Methods and Practices in Psychological Science, 3, 24-52. PsyArXiv Preprint

Frank, M. C., Bergelson, E., Bergmann, C., Cristia, A., Floccia, C., Gervain, J., Hamlin, J. K., Hannon, E. E., Kline, M., Levelt, C., Lew-Williams, C., Nazzi, T., Panneton, R., Rabagliati, H., Soderstrom, M., Sullivan, J., Waxman, S., Yurovsky, D. (2017). A collaborative approach to infant research: Promoting reproducibility, best practices, and theory-building. Infancy, 22, 421-435. PsyArXiv Preprint

*Reviewer #1:*

This paper provides two important benefits to the field. First is the clear data showing the risk of bias assessments and effect size differences between single and multi-lab studies. This provides more clear evidence about the problematic process of single study science. Second, and I believe more important, is the qualitative information provided through the interviews with participating labs that describe the barriers and recommendations from participating labs (eg timelines, pilot periods, budget limitations set by the least well-resourced lab, ethics approval processes, centralized data processing, and even interpersonal lab coordination suggestions). Summarizing these qualitative findings in specific recommendations or a set of proposed guidelines would expand the relevance of this paper.

Below are my recommendations for improvement, which are not substantial but will hopefully help the authors widen their audience and convince their peers that these methods are not limited to their own community.

Introduction: I believe that your audience reach would be expanded, and your intended audience would be even more interested in your work, by noting how this method of multi-lab collaborations is appropriate beyond preclinical research. I will not suggest changing your inclusion criteria for the study already conducted, but it would be worth noting how efforts in social psychology and human development have seen similar results as those you describe. Likewise, existing networks in psychology (https://psysciacc.org/), development biology (https://manybabies.github.io/), and special education research (https://edresearchaccelerator.org/) would be worth mentioning as models for emulation when discussing how these practices could be scaled to more communities within preclinical research (perhaps mention this in the Conclusion).

Line 123: Can you please define "interventional" and clarify your definition of "Preclinical" so as to make it clear to readers outside of your domain of focus why psychological or developmental biology was not included (as these do have human health implications). (Just to be clear, I am not being sarcastic or suggesting that these should have been included, I just know that these terms get blurry when going from the preclinical to the basic sciences).

I appreciate that the calculations for the effect size ratio and the difference of standardized mean difference were specified a priori. Even though they were straightforward calculations, there is room for divergence or selective reporting and it appears that those analyses were reported as pre-specified.

*Reviewer #2:*

This paper reports results of a systematic review comparing single-lab with multi-lab studies with regard to reported effect size and risk of bias. The main outcome was that risk of bias was lower in multi-lab studies and so was the reported effect size. The authors argue, that the higher effect sizes observed in the single lab studies are presumably over-estimates and that, hence, the multi-lab studies provide more accurate effect size estimates. They, thus, conclude that this meta-analysis demonstrates the potential value of multi-centre studies in pre-clinical research. While I do agree that multi-centre studies are valuable, I have certain reservations as to what extent this is actually shown in the current study. Here are my concerns.

1) The sample size is rather limited. While I do not doubt that the authors invested a lot of work into finding multi-centre studies fulfilling their inclusion criteria, they ended up with a list of only 16 studies. Multi-centre studies are tedious to perform and more expensive, so it is not surprising that the authors could not find more. However, given an N=16 for a rather diverse collection of studies, spanning 35 years, different model species (mice, rats, dogs, pigs, and rabbits) and investigating rather different interventions, this might be better regarded as a preliminary analysis.

2) There is one question that cannot be answered but which is nevertheless rather important. Is the difference in effect size estimates (with the multi-lab studies presumably providing more accurate effect size estimates) due to the multi-lab setting itself or due to the fact that researchers willing to go the extra length of conducting a multi-lab study are stricter and more rigorous (and more aware) than the average researcher? If the latter is the case, then this would be a confounding factor and one could not conclude that the observed difference is due to the multi-lab approach itself.

3) As the authors noted themselves, sample sizes of single-lab and multi-lab differed substantially (median 19 vs. 111) and this difference in sample size might itself lead to more accurate effect size estimates (closer to the meta-analytic estimate). I am, therefore, missing an analysis, of whether this effect alone could explain the discrepancy. That is-would single-lab studies using the same number of animals be equally accurate as a multi-lab study?

4) I have some minor methodological issues regarding the evaluation of the risk of bias and specifically its interpretation as a measure for the quality of scientific rigor. This is more of a general critique of the use of 'risk of bias' in systematic reviews and is not specific to this study, but it nevertheless means that the results of this part of the study should be taken with a grain of salt and be interpreted with care. The risk of bias questionnaire is focusing on what is reported in the publication. As it is visible in table 3, even for the multi-lab studies – where the risk of bias was considerably lower than for the single-lab studies-a substantial proportion of questions was answered with unclear – i.e., it was, for example, not reported whether allocation concealment was done or not. This could mean that it was not done or it can mean that it was done, but researchers didn't find it noteworthy to report it. Reporting practice slightly differs between fields, it differs between journals (due to space restriction, editorial guidelines, and also the rigor of the editors), and it very strongly changes over time – due to the current discussion about replicability editors and reviewers are now asking for more details than they did a decade ago. So, any analysis of the risk of bias using those criteria should consider journal and year of publication as co-variates. In the present case the authors mentioned, that they tried to match single-lab studies with multi-lab studies also according to age, though a quick look at publication years suggests that this was not always achieved.

Overall, I think this study provides evidence that multi-centric preclinical studies do report effect size estimates that might be more trustworthy than those provided by single-lab studies. I can therefore fully agree with the authors' conclusion about the value of such studies. Yet, why exactly multi-center studies fare better (whether it is because of the larger sample sizes, the different mindset of the experimenters, or the necessity to coordinate protocols), remains an open question.

*Reviewer #3:*

This is a well-written and clear manuscript covering an important and emerging area, albeit one that has not yet significantly penetrated into scientific practice. The authors were attempting to ascertain professional attitudes to multi-centre studies in pre-clinical biomedical research by using a semi-structured interview method. I am not an expert in the methods used and defer to expert reviewers in this regard; my expertise is in multi-centre animal studies.

A potential limitation is that only 16 participants were included in the interviews, although this is a sample size justified by reference to norms in the literature. Furthermore, in the results, the authors suggest that data saturation had been reached by the 13th interview, so perhaps this is sufficient. Ten out of 16 interviewees were US-based, the remainder European. A further potential bias may be that the interviewees had all participated in pre-clinical multicentre research, so were perhaps likely to be enthusiasts of the approach.

I was curious as to why the group considered that testing of efficacy was the major use of such studies (rather than testing a new mechanism or fundamental finding) and that evaluation of a new method, model, or outcome measure was not even mentioned. That would not have been my view had I been interviewed. It would be nice to see a critical discussion about this.

I wonder if a more forensic discussion of what the authors see as the probable barriers to funders, peer scientists, and other stakeholders would be informative. Perhaps even a more extensive discussion of potential solutions may have been informative.

The main area of disagreement between interviewees was the issue of harmonisation of protocols – a strong contrast to the clinical multi-centre practice where harmonisation is the norm. This interesting response could perhaps have been discussed more?

[Editors' note: further revisions were suggested prior to acceptance, as described below.]

Thank you for resubmitting your work entitled "A systematic assessment of preclinical multicenter studies and a comparison to single-laboratory studies" for further consideration by *eLife*. Your revised article has been evaluated by Mone Zaidi (Senior Editor) and a Reviewing Editor.

The manuscript has been improved but there are some remaining issues that need to be addressed, as outlined below:

*Reviewer #1 (Recommendations for the authors):*

The authors have sufficiently addressed the comments I made in the first submission.

However, one of their additions in response to another reviewer's comments may not be sufficient. The question is "Is the difference in effect size estimates (with the multi-lab studies presumably providing more accurate effect size estimates) due to the multi-lab setting itself or due to the fact that researchers are willing to go the extra length of conducting a multi-lab study are stricter and more rigorous (and more aware) than the average researcher?" I agree that the cause and effect cannot be addressed by this review and that only an experimental manipulation could start to truly answer that question (which would be a very tough experiment). However, the authors' response is also a bit limited and does add some additional questions from me. They state: "For ten multilaboratory studies, researchers had authored at least one of the matched single lab studies." I am now curious as to whether or not those ~10 single lab studies were in some way different than the ~90 other single lab studies (which would suggest that it's the person instead of the process that makes the difference). HOWEVER- I do not suggest that the authors go down that rabbit hole. No matter what is found from such digging, the root question ("is it the multi-lab process or the better scientist that makes the better studies?") will not be well addressed through this process simply because it will be an exercise in chasing smaller and smaller sub-samples with too much noise to make any reasonable conclusion. Instead, I have to disagree with the authors' first response to this comment and rather just recommend that they note these two possible explanations and state that this observational study cannot determine if it is the method or the authors that explain the differences reported.

All other changes and comments seem appropriate to me.

*Reviewer #2 (Recommendations for the authors):*

Overall the authors answered the questions coming up during the first review and changed the text accordingly.

There is, however, one point, where I would love to see a bit more. This regards the rather central question of whether multi-lab (ML) and single-lab (SL) differ because of the procedure or because they were done by a different 'population' of authors or with a different objective in mind.

The authors responded:

"We found that for the majority of the multilaboratory studies (i.e. ten), researchers had previously authored single lab studies on the same topic. " and then " This has now been addressed in the manuscript. Results: "For ten multilaboratory studies, researchers had authored at least one of the matched single lab studies." "

I would ask the authors, not only to mention in the results that for 10 ML studies, but researchers had also authored SL studies but also to describe how those 10 pairs compared. I don't advocate a statistical analysis, however, a descriptive summary (in how many of those 10 cases were the ML closer to the meta-analytic outcome than the SL?) and maybe these 10 SL studies could also be highlighted in figure 3 (if they were included there, though I presume they were).

---

## [Author Response]

Essential revisions:1) Introduction: I believe that your audience reach would be expanded, and your intended audience would be even more interested in your work, by noting how this method of multi-lab collaborations is appropriate beyond preclinical research. I will not suggest changing your inclusion criteria for the study already conducted, but it would be worth noting how efforts in social psychology* and human development** have seen similar results as those you describe. Likewise, existing networks in psychology (https://psysciacc.org/), development biology (https://manybabies.github.io/), and special education research (https://edresearchaccelerator.org/) would be worth mentioning as models for emulation when discussing how these practices could be scaled to more communities within preclinical research (perhaps mention this in the Conclusion).

These are excellent points we have now addressed in our introduction:

“…This approach has been adopted in other fields such social and developmental psychology (12, 13).”

In addition, we have mentioned the influential networks in our Discussion. We hope this will allow interested readers to seek out additional information easily.

“…. As research groups globally consider adopting this approach, the biomedical community may benefit by emulating successful existing networks of multicenter studies in social psychology (https://psysciacc.org/), developmental psychology (https://manybabies.github.io) and special education research (https://edresearchaccelerator.org/).”

2) Line 123: Can you please define "interventional" and clarify your definition of "Preclinical" so as to make it clear to readers outside of your domain of focus why psychological or developmental biology was not included (as these do have human health implications). (Just to be clear, I am not being sarcastic or suggesting that these should have been included, I just know that these terms get blurry when going from the preclinical to the basic sciences).

We agree these terms have various meanings depending on field of study. They are often a source of debate even with our closest collaborators as there are no consensus definitions. We have now clarified our intentions with these terms in the first paragraph of the methods:

Preclinical multilaboratory eligibility criteria

Population – The population of interest was preclinical, interventional, multilaboratory, controlled comparison studies. Preclinical was defined as research conducted using nonhuman models that involves the evaluation of potential therapeutic interventions of relevance to human health…

Intervention, comparators, outcomes – Interventions were restricted to agents with potential effects when considering human health.

3) I appreciate that the calculations for the effect size ratio and the difference of standardized mean difference were specified a priori. Even though they were straightforward calculations, there is room for divergence or selective reporting and it appears that those analyses were reported as pre-specified.

Thank-you for providing this supportive comment.

4) The sample size is rather limited. While I do not doubt that the authors invested a lot of work into finding multi-centre studies fulfilling their inclusion criteria, they ended up with a list of only 16 studies. Multi-centre studies are tedious to perform and more expensive, so it is not surprising that the authors could not find more. However, given an N=16 for a rather diverse collection of studies, spanning 35 years, different model species (mice, rats, dogs, pigs, and rabbits) and investigating rather different interventions, this might be better regarded as a preliminary analysis.

We would point out that, although the publication dates of included studies spanned 35 years, the majority have been published since 2015. In addition, despite the diversity of studies noted by reviewers, the overall trends are similar across studies (more complete reporting, lower risk of bias, and much smaller effect sizes than those seen in single-laboratory studies). Finally, we would note that our comparative approach emulates similar landmark clinical meta-research studies; the first highly cited landmark study on this issue in clinical research investigated only five topics (Ref #9, Bellomo 2009). Nonetheless, we agree with the reviewers that we have much to learn still and that our study is an important starting point. In our discussion we have now explicitly stated this:

“Strengths and limitations

… In addition, our quantitative analysis included only sixteen studies, and thus our results might be better regarded as a preliminary analysis that will require future confirmation when more multilaboratory studies have been conducted. We would note, however, that despite the diversity of included multilaboratory studies, overall trends were quite similar across these sixteen studies (e.g. more complete reporting, lower risk of bias, and smaller effect size than comparable single-laboratory studies).”

5) There is one question that cannot be answered but which is nevertheless rather important. Is the difference in effect size estimates (with the multi-lab studies presumably providing more accurate effect size estimates) due to the multi-lab setting itself or due to the fact that researchers willing to go the extra length of conducting a multi-lab study are stricter and more rigorous (and more aware) than the average researcher? If the latter is the case, then this would be a confounding factor and one could not conclude that the observed difference is due to the multi-lab approach itself.

In order to address this interesting question raised by the reviewers we compared authorship of the multilaboratory studies to matched single lab studies. We found that for the majority of the multilaboratory studies (i.e. ten), researchers had previously authored single lab studies on the same topic. This suggests that the multilaboratory approach itself is perhaps more rigorous.

Of interest, we would speculate that may in part due to the shared mental models that may develop in the multilaboratory approach. Of note, this is actually the topic of an ongoing funded study by our group (published protocol: DOI: 10.1371/journal.pone.0273077).

This has now been addressed in the manuscript.

Results:

“For ten multilaboratory studies, researchers had authored at least one of the matched single lab studies.”

6) As the authors noted themselves, sample sizes of single-lab and multi-lab differed substantially (median 19 vs. 111) and this difference in sample size might itself lead to more accurate effect size estimates (closer to the meta-analytic estimate). I am, therefore, missing an analysis, of whether this effect alone could explain the discrepancy. That is – would single-lab studies using the same number of animals be equally accurate as a multi-lab study?

We agree that larger sample size would lead to a more accurate effect size estimate, with smaller confidence intervals. However, as shown in Figure S12, the majority of the multi-laboratory studies effect sizes fall outside the upper limits of the 95% confidence interval of the overall single-lab studies effect size. Simply increasing the sample size would only increase precision of the estimates, but we would still anticipate the effects would fall into the confidence interval 95% of the time.

This indicates that the true value of the overall single-lab effect size does not overlap with the true effect size of multi-laboratory studies in a majority of cases. We have now mentioned this issue in the Results:

**“…**A scatterplot of the study effect sizes for each comparison is presented in Figure 3; and the forest plots of each of the 14 comparisons can be found in the supplemental information (S12 Figure). Of note, 8 of the of the 14 multilaboraotry studies had 95% confidence intervals that fell outside of the pooled single-laboratory 95% confidence intervals.”

7) I have some minor methodological issues regarding the evaluation of the risk of bias and specifically its interpretation as a measure for the quality of scientific rigor. This is more of a general critique of the use of 'risk of bias' in systematic reviews and is not specific to this study, but it nevertheless means that the results of this part of the study should be taken with a grain of salt and be interpreted with care. The risk of bias questionnaire is focusing on what is reported in the publication. As it is visible in table 3, even for the multi-lab studies-where risk of bias was considerably lower than for the single-lab studies-a substantial proportion of questions was answered with unclear-i.e. it was, for example, not reported whether allocation concealment was done or not. This could mean that it was not done or it can mean that it was done, but researchers didn't find it noteworthy to report it. Reporting practice slightly differs between fields, it differs between journals (due to space restriction, editorial guidelines, and also the rigor of the editors), and it very strongly changes over time – due to the current discussion about replicability editors and reviewers are now asking for more details than they did a decade ago. So, any analysis of the risk of bias using those criteria should consider journal and year of publication as co-variates. In the present case the authors mentioned, that they tried to match single-lab studies with multi-lab studies also according to age, though a quick look at publication years suggests that this was not always achieved.

This is a valid point raised by the reviewers. We have now acknowledged this in the Discussion:

“…Another limitation is that our assessment of risk of bias relies on complete reporting; reporting, however, can be influenced by space restrictions in some journals, temporal trends (e.g. better reporting in more recent studies), as well as accepted norms in certain fields of basic science. However, with the increasing prevalence of reporting checklists and standards in preclinical research, (62) future assessments will be less susceptible to this information bias (63).”

8) There seem to be some inconsistencies between Table 3 and Table S11. For example, in the column "Blinding of outcome assessment" in Table S3 there are 4 studies with "unknown" (Crabbe, Alam, Spoerke, Gill), while in Table S11 in the column "Blinding of outcome assessment" there are 4 study scoring 0, but here Arroyo-Araujo scores also 0, while Gill scores 1. How does this fit together?

Thank you for pointing this out – the inconsistencies were noted and have been corrected. Studies Gill and Arroyo-Araujo both blinded the outcome assessment, thus, in Table 3 the “unknown” was changed to “low” for Gill, 2016 under "Blinding of outcome assessment"; and in Table S10 the “0” was changed to “1” for Arroyo-Araujo, 2019 under "Blinding of outcome assessment". Additionally, the Results section has been adjusted accordingly – changed from “Twelve studies” to “Thirteen studies”.

9) Methods: maybe I have missed that: have you described, whether for the reporting scores you checked only the main article or also the electronic supplementary (supporting information)?

Indeed, we did consult the supporting information in addition to the main article for evaluating the reporting scores. This has now been clarified in the methods:

“For both assessments, the main articles along with the supporting information (when provided) were consulted.”

10) Language limits: In the methods description you wrote "No language limits were imposed" (line 147). In the Results section, I could, however, not find any information, on if any non-English papers were found and whether they were included in the analysis. There is also no mention of what you would have done if you encountered a paper in a language you were not capable of. When would this have been an exclusion criterion? In any case, in the methods section where you wrote "no language limits were imposed", please add the information that all search terms were in English. (Or were they also translated into other languages?)

We did not impose any language limits in the search and no non-English articles were found. For other meta-research projects we have typically had the article translated by members in our research institute familiar with the language.

We have added the following to clarify in the Results:

“There were no non-English articles identified in the search.”

11) Line 166: Primary and secondary research outcomes. According to my own experience, one sees many papers, where the authors report outcomes of several analyses and do not clearly define what the primary and the secondary outcomes are. In fact, I only rarely read that an author writes "this is our primary outcome" and "this was a secondary outcome". Did you encounter such cases and which criteria did you use to decide what counted as a primary and secondary outcome?

This is an important point and reflects differences in approach and language used by lab researchers (versus clinical trialists, for instance). Thus, we have now further clarified in the Methods:

“Qualitative data included… all study outcomes (primary, secondary, or undefined) …” and “Quantitative data included… sample sizes for the outcome used in the meta-analysis and for the control group.”

The heading in Table 2 was changed from “Primary Outcome” to “Study Outcomes”.

12) Line 196: Rapid review method. Please describe in one or two sentences what that is.

Thank you for this suggestion. We added a more detailed description to the Methods:

“…, which consisted of the search of a single database, and by having a single reviewer screen, extract, and appraise studies while an additional reviewer auditing all information.”

13) Line 259: were the effect sizes just pooled or were they weighted by study size or variance?

Pooled effect sizes (i.e. for single lab studies) were weighted (inverse variance).

For the effect size ratio calculation, there was no weighting of the two data points used (i.e. multilab effect size, pooled single lab effect size). We recognized a priori the potential drawbacks of this, and this is why several measures to compare effect sizes of multilaboratory vs. single lab studies were prespecified and reported.

14) Line 267: I think giving a ratio (without multiplication by 100) might be better

Point taken. We have modified the effect size ratio calculation to be a simple ratio without multiplying by 100%. The has been adjusted in the methods, results, and supporting information Table S11.

15) Line 345: potentially industry-related influences are considered a risk of bias. Is there actually any study clearly showing that papers with authors from industry are more biased than papers from authors from academia? I know that this is an item on the SYRCLE list of bias tools, but is this prejudice against industry justified – i.e., based on evidence? Just recently there have been several cases of spectacular fraud (data fabrication) by authors in academia (behavioural ecology). Specifically for early career researchers in academia publication pressure for securing tenure or a faculty position is often described as extremely high, creating incentives for "beautifying" results which are then easier to publish. If you are, however, aware of any publication or data showing that bias in industry is more of an issue than in academia then it would be good to cite it here.

This is an important point. Dr. Lisa Bero and colleagues have investigated this issue in several studies and demonstrated that the relationship is complicated. For studies investigating statins on atherosclerosis and bone outcomes they found that non-industry studies were more likely to demonstrate efficacy than industry sponsored studies (i.e. larger effect size in non-industry studies; PMID 25880564 and 24465178). However, in animal studies assessing the effect of atrazine exposure, industry sponsored studies were less likely to demonstrate harm than non-industry sponsored studies (i.e. smaller effect size in non-industry studies; PMID 26694022). In our group’s work (not-published yet) we found that industry sponsored animal studies of hydroxyethyl starch intravenous fluids were more likely to demonstrate efficacy than non-industry sponsored studies (i.e. smaller effect size in non-industry studies). This is notable since these (very expensive) fluids were subsequently pulled from markets with a black-box warning from the FDA.

These data demonstrate that there is no straightforward/easy approach to understanding the relationship between industry/non-industry sponsorship and risk of bias in preclinical studies.

We agree that perhaps the SYRCLE risk of bias tool should reevaluate this issue in the future and we have brought it to their attention through an email communication with Dr. Kimberly Wever, who is a member of their leadership. Ultimately, this remains a question that needs to be investigated further and addressed by future iterations of the SYRCLE risk of bias tool.

16) Line 388: As mentioned in the public review: I would be very careful with equating low reporting scores with "quality of research"

We agree and have adjusted our terminology throughout (i.e. “completeness of reporting”).

17) Analysis (Meta-analysis, Figure S12): It would be really interesting to add publication year as a covariate in the models. I would assume that estimates get closer to zero the younger the publication.

In order to address this question we have explored the relationship between absolute effect size vs. year of publication (all single lab studies plotted in Author response image 1). There is no clear relationship between the two variables (r^2^ = 0.002).

**Author response image 1. sa2fig1:** 

18) The list of references seems incomplete or incorrect with errors in text citation numbers. There are far more citations in the text (up to 82, whereas only 17 are on the list).

Thanking for pointing this out. There was an error in the continuation of the references/citations for the interview study that was appended. This has now been corrected.

19) It would be good to know more about the training/qualification/validation criteria for the interviewers.

This is an important issue in qualitative research and reflects on reflexivity. As such, it is a mandatory aspect of the COREQ reporting guideline we used (found in the appendix for that paper).

20) Page 3 the sentence starting "we found that the…..rigorous manner" did not quite make sense to me.

Thank you for pointing this out in the appended interview study. We have adjusted the sentence to, “We found that the multicenter studies undertaken to date have been conducted using more rigorous methodology and quality control”.

21) I was curious as to why the group considered that testing of efficacy was the major use of such studies (rather than testing a new mechanism or fundamental finding) and that evaluation of a new method, model, or outcome measure was not even mentioned. That would not have been my view had I been interviewed. It would be nice to see a critical discussion about this.

Indeed, we agree that multilaboratory studies could be used for other purposes (e.g. mechanistic/fundamental). However, the majority of studies using this method have been interventional/therapeutic. We have now discussed this in the Discussion:

Discussion:

“…The application of rigorous inclusion criteria limited the eligible studies to interventional, controlled-comparison studies, which could omit valuable information that may have come from the excluded studies of non-controlled and/or observational designs, or studies focused on mechanistic insights.”

22) I wonder if a more forensic discussion of what the authors see as the probable barriers to funders, peer scientists, and other stakeholders would be informative. Perhaps even a more extensive discussion of potential solutions may have been informative.

As we had only interviewed the scientists involved, and not other stakeholders, we are hesitant to speculate on solutions from their viewpoints. Although we are hesitant to further lengthen our discussion (which we feel is already quite long), we have now included a statement that further studies will need to investigate this important issue. We believe a stakeholder response should be:

Discussion:

“…Moreover, identified barriers and enablers to these studies should be further explored from a variety of stakeholder perspectives (e.g. researchers, animal ethics committees, institutes and funders) in order to maximize future chances of success.”

23) The main area of disagreement between interviewees was the issue of harmonisation of protocols- a strong contrast to the clinical multi-centre practice where harmonisation is the norm. This interesting response could perhaps have been discussed more?

This is an excellent point and we have now elaborated on this issue in the Results:

“If the protocol was to be harmonized, then it had to be adapted to fit each lab’s budget accordingly (i.e. the lab with the smallest budget set the spending limit (19)). Alternatively, labs developed a general protocol but adapted it to fit their own respective budget with what resources they had. Of note, recent work has suggested that harmonization reduces between-lab variability, however, systematic heterogenization did not reduce variability further (52); this may suggest that, even in fully harmonized protocols, enough uncontrolled heterogeneity exists that further purposeful heterogenization has little effect.”

[Editors' note: further revisions were suggested prior to acceptance, as described below.]

Reviewer #2 (Recommendations for the authors):Overall the authors answered the questions coming up during the first review and changed the text accordingly.There is, however, one point, where I would love to see a bit more. This regards the rather central question of whether multi-lab (ML) and single-lab (SL) differ because of the procedure or because they were done by a different 'population' of authors or with a different objective in mind.The authors responded:"We found that for the majority of the multilaboratory studies (i.e. ten), researchers had previously authored single lab studies on the same topic. " and then " This has now been addressed in the manuscript. Results: "For ten multilaboratory studies, researchers had authored at least one of the matched single lab studies." "I would ask the authors, not only to mention in the results that for 10 ML studies, but researchers had also authored SL studies but also to describe how those 10 pairs compared. I don't advocate a statistical analysis, however, a descriptive summary (in how many of those 10 cases were the ML closer to the meta-analytic outcome than the SL?) and maybe these 10 SL studies could also be highlighted in figure 3 (if they were included there, though I presume they were).

The following text has been added:

Methods:

“Peer reviewers requested additional analysis to qualitatively assess identified single-laboratory studies that were conducted by the same authors of matched multilaboratory studies.”

Results:

“The ESR was greater than 1 (i.e. the single lab studies produced a larger summary effect size) in 10 of the 14 comparisons. The median effect size ratio between multilaboratory and single lab studies across all 14 comparisons was 4.18 (range: 0.57 to 17.14). The ESRs for each comparison along with the mean effect sizes and ratio of the mean effect sizes is found in the supplemental information (Appendix 11 – table 7).

For 10 multilaboratory studies, researchers also had authored 11 matched single lab studies. Median effect size ratio of this smaller matched cohort was 2.50 (mean 4.10, range 0.35-17.63). Median quality assessment score for the 10 multilaboratory studies was 3 (range 1-5); median quality score of the 11 single-laboratory studies matched by authors was 1 (range 1-4).”

Discussion:

“The difference in observed methodological quality and high completeness of reporting in preclinical multilaboratory studies could be explained by the routine oversight and quality control that were employed by some of the multilaboratory studies included in our sample. Though not all multilaboratory studies reported routine oversight, we expect that this is inherent of collaborative studies between multiple independent research groups. As reported by several studies, the coordination of a successful preclinical multilaboratory study requires greater training, standardization of protocols, and study-level management when compared to a preclinical study within a single-laboratory. Another barrier was the issue of obtaining adequate funding for a multilaboratory study. As consequence of limited funding, we would speculate that these studies may have undergone more scrutiny and refinement by multiple investigators, funders, and other stakeholders Indeed, comparison single-laboratory studies that had been conducted by authors of multilaboratory studies suggested differences in the conduct and outcomes of these studies (despite having the some of the same researchers involved in both). However, this post hoc analysis was qualitative with a limited sample; thus, future studies will need to explore these issues further.”

We chose not to highlight the studies in Figure 3 to avoid distracting from our a priori planned analysis.